# Soil microbial responses to multiple global change factors as assessed by metagenomics

Álvaro Rodríguez del Río [1] ✉, Stefan Scheu [2,3] & Matthias C. Rillig [1,4]

Anthropogenic activities impose multiple concurrent pressures on soils globally, but responses of soil microbes to multiple global change factors are poorly understood. Here, we apply 10 treatments (warming, drought, nitrogen deposition, salinity, heavy metal, microplastics, antibiotics, fungicides, herbicides and insecticides) individually and in combinations of 8 factors to soil samples, and monitor their bacterial and viral composition by metagenomic analysis. We recover 742 mostly unknown bacterial and 1865 viral Metagenome-Assembled Genomes (MAGs), and leverage them to describe microbial populations under different treatment conditions. The application of multiple factors selects for prokaryotic and viral communities different from any individual factor, favouring the proliferation of potentially pathogenic mycobacteria and novel phages, which apparently play a role in shaping prokaryote communities. We also build a 25 M gene catalog to show that multiple factors select for metabolically diverse, sessile and non-biofilm-forming bacteria with a high load of antibiotic resistance genes. Finally, we show that novel genes are relevant for understanding microbial response to global change. Our study indicates that multiple factors impose selective pressures on soil prokaryotes and viruses not observed at the individual factor level, and emphasizes the need of studying the effect of concurrent global change treatments.

Human pressures are numerous, highly diverse in nature[1], and influence soil ecosystems at a global scale. One of the most important effects of global change (GC) are shifts in soil microbial populations, central to soil functioning. Several experiments have revealed the response of soil biota to alternative GC factors like warming[2], drought[3] or microplastics[4], among others[5]. These microbial disturbances impact important soil functions, and monitoring them remains relevant for understanding anthropogenic impacts on soil ecosystems.

However, most studies only include a limited number of GC factors, even though many may act concurrently in natural conditions. In order to address this gap, Rillig et al.(2019)[6] designed a multifactor experiment including 10 GC factors of diverse nature[1]: warming (physical factor), drought, nitrogen deposition, increased salinity and heavy

metal (inorganic chemical factors), microplastics (particle contamination) and antibiotics, fungicides, herbicides and insecticides (organic chemical toxicants). After applying them individually and in an increasing number of simultaneous combinations (up to 10) to soil grassland samples, results showed that multiple concurrent GC factors triggered directional shifts in soil properties. For instance, individual GC factors barely affected water drop penetration time, but the application of multiple concurrent factors caused a significant increase, more pronounced as the number of applied factors increased. These results highlighted the importance of studying not only the effect of individual GC factors, but also the combined effect of many.

Increasing GC factors also triggered directional changes on soil fungal populations, but whether other microorganisms follow the

[1]Institute of Biology, Freie Universität Berlin, Berlin, Germany. [2]JFB Institute of Zoology and Anthropology, University of Göttingen, Göttingen, Germany. [3]Centre of Biodiversity and Sustainable Land Use, University of Göttingen, Göttingen, Germany. [4]Berlin-Brandenburg Institute of Advanced Biodiversity Research (BBIB), Berlin, Germany. ✉e-mail: alvarordr94@gmail.com

same patterns remains unknown. This includes prokaryotes, which usually show different dynamics than fungi[7,8] and are central to soil functioning. For instance, they are the only fixers of molecular nitrogen, a limiting soil nutrient, mediate phosphorus mobilization, critical for plant growth, decompose plant derived organic matter, and contribute to soil structure through the formation of aggregates[9–11]. Additionally, the response of viruses, understudied players of soil functioning with a key role in regulating microbial host dynamics and soil carbon pools[12], to multiple GC factors has also not been studied.

In order to understand whether prokaryotes and viruses show different responses to multiple GC factors than to individual GC treatments, we leverage 70 samples from the multi-factor experiment by Rillig et al (2019)[6], including i) 10 controls, ii) 50 single GC factor samples (5 samples treated with each individual factor), and iii) 10 samples treated with random combinations of 8 concurrent GC factors (see Supplementary Fig. 1 for the factor composition of the samples), and analyse them following a comprehensive metagenomic exploration (Fig. 1). The 8-factor treatment de-emphasizes (through the random draws) the composition of factors, represents the changing multifactorial conditions observed in nature[13], and yielded unexpected patterns in the original study by Rillig et al. (2019), making it appropriate for studying the effect of multiple concurrent GC factors.

## Results

### Different prokaryotic composition under alternative GC scenarios

Soils are the most biodiverse habitat on the planet[14], and contain an immense number of uncultivated microbial species not present in reference databases[15]. These unknown species can be uncovered by constructing Metagenome-Assembled Genomes (MAGs) de novo, a method that is revealing a great degree of unknown biodiversity[16]. Hence, after sequencing the metagenomes of the 70 soil samples from the experiment by Rillig et al. (2019), trimming the reads and assembling them into contigs, we aimed at binning them into prokaryotic MAGs.

After comparing the performance of different prokaryotic binning strategies (Supplementary Table 1), we restricted our analysis to the genomic bins computed with the multi-sample binning strategy by SemiBin2[17], which provided 653 medium quality (completeness ≥ 50%, contamination <10%) and 89 high quality (completeness ≥ 90%, contamination <5%) genomic bins[18]. These MAGs show variable genome sizes across treatments: genomic bins reconstructed from samples treated with heavy metal and the random combination of 8 GC factors are significantly larger than in the control samples (Two-sided Wilcoxon test $p = 8.5e\text{-}06$ and 0.004, respectively; Fig. 2C, Supplementary Fig. 2).

MAGs across samples were redundant and clustered into 77 reference species bins according to the 95% Average Nucleotide Identity (ANI) species definition. Even though we could resolve the phylum, class and order of all, 96.1% reference MAGs could not be taxonomically classified to the species level with GTDB-tk2[19], indicating they represent unknown taxa. For simplicity, we refer to bins not assigned species-level taxonomic labels as unknown.

Even though MAGs usually gather the most abundant species, the 77 representative MAGs captured less than 10% of the metagenomic reads, indicating they represent an incomplete picture of biodiversity. This is typical of de novo genome building[20,21], especially in complex environments like soil[16,22]. Also, given that MAGs were directly reconstructed from samples exposed to the different treatments, changes in composition could arise because of a bias in the reference sequences used. Hence, we did not only rely on the collection of MAGs for testing prokaryotic community patterns, and calculated taxonomic profiles with other profiling methods. As many reference-based strategies with different strengths and limitations exist[22], we tested three taxonomic profiling tools following two different approaches: marker gene detection (mOTUs[23] and SingleM[15]) and sequencing read classification based on k-mer frequencies (Kraken2[24]).

We then measured the overlap between the taxa identified by MAGs and by reference-based methods. Many prevalent taxa located by singleM are absent in the MAG collection, including several archaeal phyla and some ubiquitous and typically abundant soil members such as *Rhodoplanes*[25]. In contrast, reference-based methods did not target members of the Bacillota phylum, from which we recovered 12 highly unknown MAGs (i.e., 4 bins from unknown families) (Supplementary Fig. 3).

We next exploited the genomic bins and reference-based taxonomic profiles for understanding the effect of the different GC treatments to prokaryotic populations. We observed clear diversity shifts across treatments (Bray-Curtis beta-diversity, ANOVA, $p = 8.8e\text{-}5$, Fig. 2A, Figs S6–8). For instance, most GC scenarios, except drought, warming and fungicides significantly reduced the relative abundance (Two-sided Wilcoxon test, $p < 0.05$, Supplementary Table 2) of a potentially nitrogen-fixing unknown MAG within the *Bradyrhizobium* genus, ubiquitous and globally abundant in soil[25].

Most individual global change factors led to significant differences in bacterial alpha diversity compared to control samples. For instance, according to the abundance profile of the genomic bins, heavy metal and salinity trigger significant diversity losses (Fig. 2B) and drive the strongest responses (34.8 and 31.1% mean decrease accuracy in a random forest regression, respectively, Supplementary Fig. 4). In contrast, other factors such as fungicide, nitrogen deposition and drought significantly increased alpha diversity. However, the random combinations of 8 GC factors, regardless of their identity, always

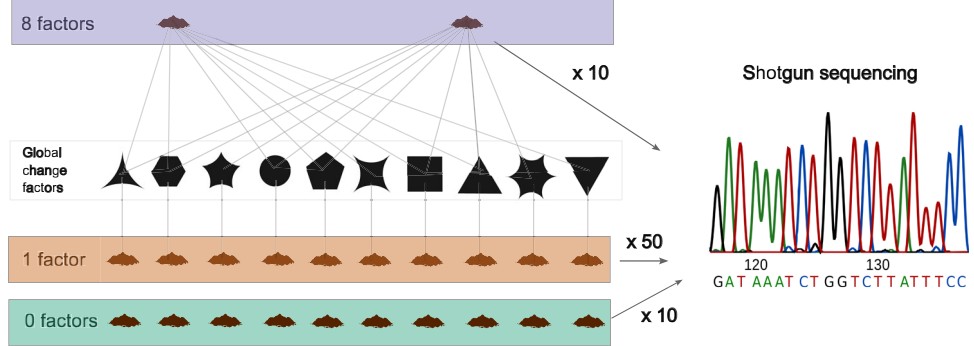

**Fig. 1 | Graphical summary of the experimental set-up.** We sequenced the metagenomes of 10 control samples (treated with 0 factors, green), 50 samples treated with 10 individual GC factors (5 samples per factor, orange) and 10 samples treated with random combinations of 8 GC factors (purple), and sequenced their metagenomes for characterizing their prokaryotic and viral microbiome.

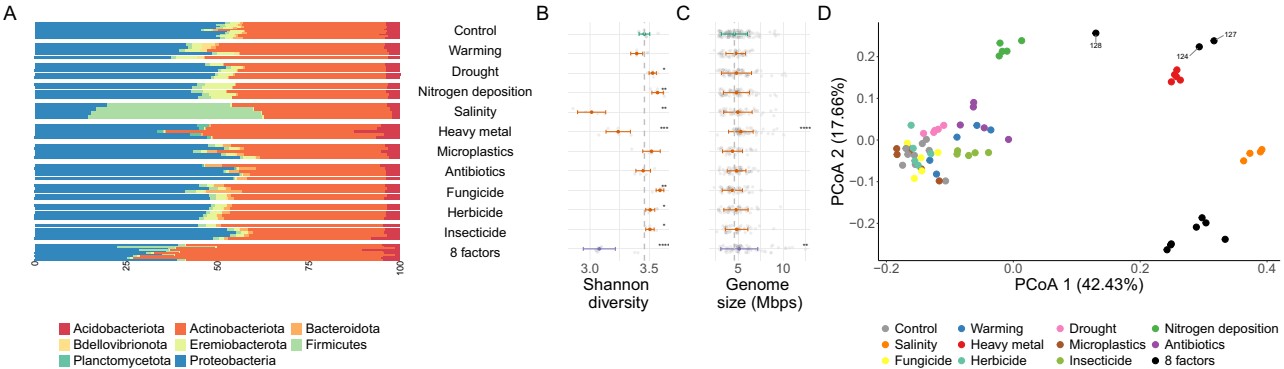

**Fig. 2 | Bacterial composition and diversity change across GC conditions.**
**A** Taxonomic profile of the representative MAGs reconstructed from the samples included in this analysis (10 controls, 5 for 10 different GC factors and 10 random combinations of 8 GC factors) collapsed to the phylum level; (**B**) Shannon diversity index, per treatment; (**C**) MAG genome size (corrected by completeness), per treatment; (**D**) Principal Coordinate Analysis (PCoA) on the relative abundance of the reference MAGs. We indicate 8-factor samples not including the salinity treatment (124, 127, 128). In (**B**, **C**), thick points represent the median values, and bars indicate standard deviation intervals. Asterisks represent different significance levels obtained after a Two-sided Wilcoxon test comparing control samples with the samples to which GC treatments were applied; * indicate $p \leq 0.05$, **$p \leq 0.01$, ***$p \leq 0.001$ and **** $\leq 0.0001$. 10 Control samples, 5 samples for each individual GC treatment, and 10 8-factor samples were considered in the statistical analyses. Exact $p$-values are provided in Supplementary Tables 3 and 4. Source data are provided as a Source Data file.

decreased bacterial alpha diversity, and the number of factors have a more important contribution in explaining diversity patterns than most individual treatments (11.2% mean decrease accuracy in a random forest regression, 5th ranked treatment, Supplementary Fig. 4). Similar patterns were observed in the taxonomic profiles by mOTUs and SingleM (alpha diversity Spearman correlation $R = 0.43$ ($p = 0.0002$) and 0.33 ($p = 0.006$), respectively, Figs. S5–7), which also report diversity losses after the heavy metal and salinity treatments. However, only mOTUs, which have been proved to perform well in estimating diversity[26], found significant diversity losses after the application of the 8-factor treatments. Diversity predictions by Kraken2, which applies a k-mer approach for species detection, are tangentially different from the other three methods (Supplementary Figs. 5,8), possibly because of its high sensitivity, which has an important impact on diversity metrics[22].

In contrast, all four methods agree on their beta-diversity estimates (Supplementary Fig. S5). Community composition was markedly different after heavy metal, nitrogen deposition, salinity and the 8-factor treatments (Fig. 2D, Figs S6–8). For instance, salinity samples are characterized by an abrupt increase in the relative abundance of Firmicutes bins (Two-sided Wilcoxon test, $p = 0.002$, Fig. 2A). The 8 concurrent factor samples have the highest intra-treatment variability, but also show common community patterns distinct to the control samples (Bray-Curtis beta-diversity, ANOVA-like pairwise permutation test, $p = 0.003$). For instance, they repeatedly show an increased abundance of Actinomycetia class genomes (Fig. 2A), driven by an unknown bin within the Chersky-822 genus, which doubles in relative abundance to become the second most abundant species (Two-sided Wilcoxon test, $p = 0.002$). Along the first PCoA axis, 8-factor samples cluster together with heavy metal and salinity samples, indicating similarities in their composition. However, they distribute differently across the second axis (which explains 17.7% variance) and form two well differentiated clusters. The first is composed of 7 samples which include the salinity treatment, and the second consists of the 3 remaining samples, which miss the salinity treatment (sample numbers 124,127 and 128, indicated in Fig. 2D). The biggest difference between these two clusters was the abundance of Actinobacteriota genomes, which increased in samples 124, 127 and 128 (Two-sided Wilcoxon test, $p = 0.007$) while decreasing in the remaining 8-factor samples (Two-sided Wilcoxon test, $p = 0.09$). The 8 GC factor samples treated with salinity also exhibit a reduced abundance of Proteobacteria (Two-sided Wilcoxon $p = 0.0001$), similarly to the individual salinity

treatment (Two-sided Wilcoxon test, $p = 0.002$), which was not observed in samples 124, 127 and 128, missing the salinity treatment (Two-sided Wilcoxon $p = 0.28$).

The distinctive community of the 8 GC factor treatment samples may be a consequence of high rates of cell death because of highly stressful conditions, which may have increased the relative abundance of resistant members even if their absolute abundance remained similar or decreased more slowly. In order to test whether this was the case, we conducted a Phospholipid Fatty Acid (PFLA) analysis, which provides estimates of alive bacterial biomass, as phospholipids from dead cells are rapidly degraded. The only treatment with significant effects was fungicide, which trigger an increase in bacterial biomass, possibly because of a decrease in the abundance of fungal competitors. The 8-factor treatment did not decrease bacterial biomass significantly (Supplementary Fig. 9), and beta-diversity estimates on absolute abundances provide similar patterns as relative species counts (Supplementary Fig. 10), indicating that other mechanisms than differential cell death across treatments are triggering the observed patterns.

### Multiple global change factors drive an increase of unknown mycobacteria

Some individual factors like salinity, heavy metal, drought and nitrogen deposition increased the relative abundance of 5 reference bins from unknown species within the *Mycobacterium* genus (Wilcoxon test, $p < 0.05$, Supplementary Table 5). This increase was especially evident after the application of 8 concurrent GC factors (Two-sided Wilcoxon test, $p = 1e-5$, Fig. 3A), where the *Mycobacterium* MAGs are among the 12 most abundant genomes, whereas in the control samples the most abundant is ranked 27th. We also observed a significant increase in mycobacterial biomass in the 8-factor samples (Two-sided Wilcoxon test $p = 1e-5$, Supplementary Fig. 11), indicating that their increment in relative abundance was not a consequence of the death of other taxa, but due to actual growth.

Although some mycobacteria have positive health effects[27], many species, known as Non-Tuberculous *Mycobacterium* (NTM) are environmental opportunistic pathogens[28], and are becoming an increasing sanitary problem[29]. In order to understand the potential pathogenicity of the unknown *Mycobacterium* MAGs thriving in multiple GC conditions, we examined their virulence factor content[30], which, despite not being markers for pathogenicity, contribute to the ability of pathogens to survive within hosts[31], and are common in mycobacteria. All

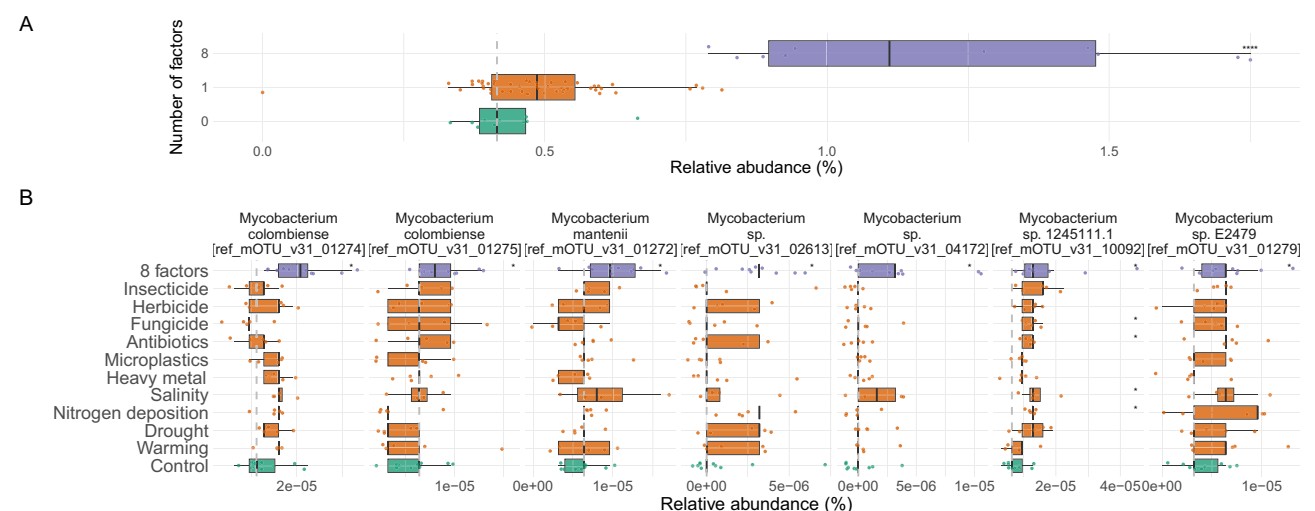

**Fig. 3 | *Mycobacterium* species are enriched in multiple GC factor samples.**
**A** Relative abundance of *Mycobacterium* bins in control, one factor ($p = 0.06$) and 8-factor ($p = 0.00001$) samples; (**B**) Abundance of the 7 *Mycobacterium* OTUs significantly enriched after the 8-factor treatment (Two-sided Wilcoxon test $p$ value < 0.05). Data are represented as boxplots in which the middle line is the median, the lower and upper hinges correspond to the first and third quartiles, the upper whisker extends from the hinge to the highest value no further than $1.5 \times$ interquartile range (IQR) from the hinge and the lower whisker extends from the hinge to the lowest value no further than $1.5 \times$ IQR of the hinge. Asterisks represent different significance levels obtained after a Two-sided Wilcoxon test comparing control samples with the samples to which GC treatments were applied; * indicate $p \leq 0.05$, **$p \leq 0.01$, ***$p \leq 0.001$ and **** $\leq 0.0001$. 10 Control samples, 5 samples for each individual GC treatment, and 10 8-factor samples were considered in the statistical analyses. Exact $p$-values for B) are provided in Supplementary Table 6. Source data are provided as a Source Data file.

unknown *Mycobacterium* bins contained a high content of virulence factor genes compared to the rest of MAGs (Supplementary Fig. 12) and similar to other soil NTM MAGs reconstructed in Bin Ma et al. (2023)[32–34] (Supplementary Fig. 13).

Among the virulence factors in the five unknown *Mycobacterium* reference bins, many are characteristic of *Mycobacterium tuberculosis*, which share genomic characteristics with NTM[35]. For instance, they all show copies of *MmpL13*, essential for the integrity of the mycobacterial envelope[36] and type VII secretion systems, which transports proteins across this outer membrane and play a significant role in mycobacterial virulence[37]. Four bins also contain *Lsr2* genes, involved in the metabolism of *Mycobacterium tuberculosis* during chronic infection[38]. Additionally, one MAG carries a phospholipase C, which contributes to phagosome escape and facilitates spread[39], and three contain copies of *MgtC*, a membrane protein that promotes survival of intracellular pathogens[40].

Only 5% of the genes classified as *Mycobacterium* were included in *Mycobacterium* bins, indicating that MAGs did not capture the whole genus diversity. Hence, we next asked whether other already described *Mycobacterium* species may also increase in abundance after the application of multiple GC factors. For this purpose, we exploited the taxonomic profiles built with mOTUs[23], which detected 55 *Mycobacterium* Operational Taxonomic Units (OTUs). Among them, seven OTUs significantly increased in abundance in the 8 GC factor samples (Two-sided Wilcoxon test, p < 0.05), including *Mycobacterium colombiense* and *Mycobacterium manitenii*, both of which already caused human infections[41,42] (Fig. 3B).

To our knowledge, NTM infections from soil have not been reported. In contrast, water is considered to be the main source of NTM human infections[43]. Using data from a previous study by Romero et al. (2020)[44], we tested whether multiple GC factors (warming and pesticides, which were applied individually and in combination in such study) increased the relative abundance of *Mycobacterium* in river biofilms. Even though we could not annotate any *Mycobacterium* Amplicon Sequence Variant (ASV) to the species level, the only mycobacterial ASV showing variability across treatments is significantly enriched after the two factors

were applied concurrently (Two-sided Wilcoxon test $p = 0.01$, Supplementary Fig. 14).

Whereas no other potential pathogen was included within the MAGs, many genera containing pathogenic species (gathered in the MBPD database[45]) were detected by alternative profiling methods. Among them, *Bacillus* and *Phychrobacillus* solely increase in abundance after the application of the 8 concurrent GC factors (Two-sided Wilcoxon test, p < 0.05), suggesting that multiple GC conditions may select for additional potentially pathogenic taxa (Supplementary Fig. 15).

## Global change factors select for unknown members of the rare biosphere

Even though all samples came from the same soil location, 49 reference bacterial bins (63.6%) did not include any genome reconstructed from the control samples after dereplication, and only became evident after applying some GC treatments. Some of these MAGs were not even detected in control samples, and showed strong increases in abundance after some GC treatments, indicating the power of sample manipulation for uncovering low-abundant taxa[46].

For instance, 5 Firmicutes genomes under the detection threshold in control samples significantly increase in abundance (Two-sided Wilcoxon test p < 0.05) and gathered more than 0.1% of the metagenomic reads after the salinity treatment. Among them, we found three unknown genus genomes within the Sporolactobacillaceae family, previously reported to have salt - tolerant members[47], one of which became the most abundant species. Additionally, two conditionally rare unknown bins are classified within the extremophile Alicyclobacillaceae family[48]. Similarly, 4 genomes mostly undetected in the control samples significantly increase in abundance (Two-sided Wilcoxon test p < 0.05) and gather more than 0.1% of the metagenomic reads after the heavy metal treatment, including two unknown MAGs from the *Edaphobacter* genus (within the extremophile Acidobacteriae class[49]), an unknown genus MAG within the Isosphaeraceae family, and an unknown 17J80-11 genus bin from the Caulobacteraceae family, which has some copper resistant members[50]. Nitrogen deposition also favoured rare genomes across the bacterial phylogeny (Supplementary Fig. 16).

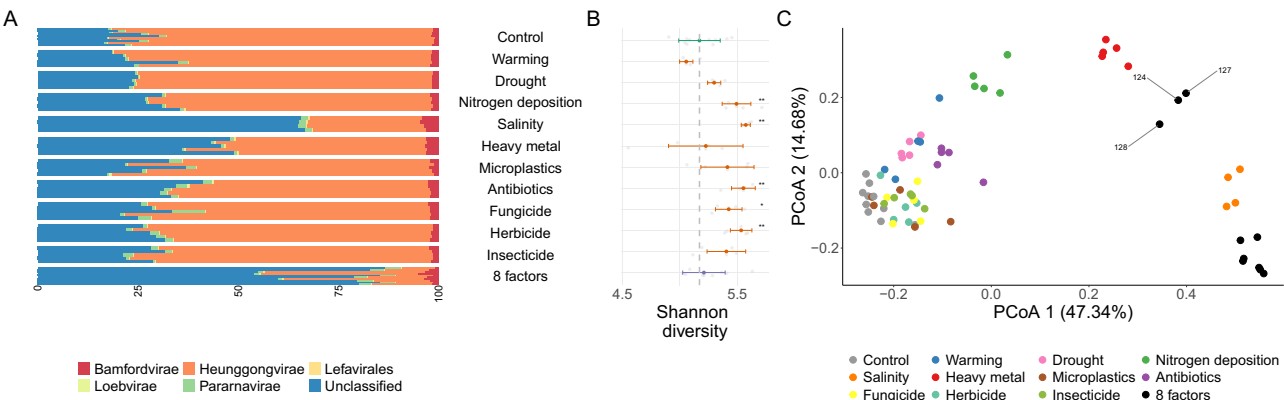

**Fig. 4 | Viral composition and diversity change across GC conditions.**
**A** Taxonomic profile of the representative viral MAGs reconstructed from the 70 samples included in this analysis (10 controls, 5 for 10 different GC factors, 10 random combinations of 8 factors) collapsed to the class level; (**B**) Shannon diversity index, per treatment. Thick points represent the median values, and bars indicate standard deviation intervals. Asterisks represent different significance levels obtained after a Two-sided Wilcoxon test comparing control samples with

the samples to which GC treatments were applied; * indicate $p \le 0.05$, **$p \le 0.01$, ***$p \le 0.001$ and **** $\le 0.0001$. 10 Control samples, 5 samples for each individual GC treatment, and 10 8-factor samples were considered in the statistical analyses. Exact $p$ values are provided in Supplementary Table 8. **C** Principal coordinate analysis based on the taxonomic annotations of the reference viral MAGs. We indicate 8-factor samples not including the salinity treatment (124, 127, 128). Source data are provided as a Source Data file.

Conditionally rare taxa are important for environmental responses to perturbations[51,52] because they serve as reservoirs of genetic diversity not necessarily encoded by more abundant taxa[53], which allow them perform better in determined conditions, and contribute to environmental resistance and resilience[53–55]. After a differential frequency analysis, we found that the 5 conditionally rare Firmutes genomes enriched after the salinity treatment show a higher frequency of many genes involved in sporulation (Two-sided Wilcoxon test with Bonferroni-adjusted $q < $1e-20, Supplementary Data 1). Similarly, the four rare genomes enhanced by heavy metal are enriched (Two-sided Wilcoxon test with Bonferroni-adjusted $q < 0.01$) in several metal transporter and succinoglycan biosynthesis sequences, which can confer resistance to extreme conditions[56], among other genes (Supplementary Data 2).

### Different global change conditions increase the abundance of unknown viruses

We next exploited the metagenomic assemblies of the 70 samples to ask whether GC conditions also select for different viral populations. For monitoring how viral abundance shifts under different GC treatments, we first identified viral contigs with VirFinder[57], and phage contigs with Seeker[58]. Both viruses and phages increase in frequency after some treatments, especially salinity, nitrogen deposition and the random combination of 8 GC factors (Supplementary Table 7, Supplementary Fig. 17). Phage relative abundance correlated with bacterial composition (Spearman $R = 0.59$, $p = $1e-7), suggesting that phages may be important players in shaping the soil microbiome when exposed to GC conditions[12]. In contrast, we found a negative but non-significant correlation between phage frequency and bacterial biomass (Spearman $R = -0.15$, $p = 0.2$), indicating that populations targeted by phages are substituted by other community members (Supplementary Fig. 18).

In order to understand which phage taxa drove these responses, we computed de novo phage bins with PHAMB[59] after MAG calculation with VAMB on the contigs assembled for each sample. We reconstructed 882, 931 and 52 medium quality, high quality and complete MAGs, respectively, which de-replicated into 895 reference viral bins. Among them, 48.5%, including 4 complete bins, do not match any reference sequence, and increased in abundance after most treatments, especially the 8 concurrent GC factors (Two-sided Wilcoxon test, $p = $1e-5, Fig. 4A). In contrast to bacteria, viral diversity increased after most treatments (Fig. 4B, Supplementary Table 8), with salinity,

warming, heavy metal and antibiotics as the most important alpha diversity determinants (mean decrease accuracy in a random forest regression higher than 18%). Compositional analyses indicated a differential viral composition under multiple GC conditions, with the 8-factor samples gathering into two well differentiated clusters (Fig. 4C), mirroring bacterial composition.

Because, similarly to prokaryotic MAGs, phage bins may be missing a substantial part of the viral diversity, we confirmed viral community patterns on the taxonomic profiles built by Kraken2. Kraken2 which, showed different alpha diversity patterns than viral MAGs, but confirmed a distinctive viral composition after the application of 8 GC factors along the first PCoA axis (Supplementary Fig. 19).

### Different life history traits under global change

Distantly related species can show similar genes and perform overlapping ecosystem functions[60]. Hence, we next asked whether the differences observed at the taxonomic level translated into different functional repertoires of the communities. For this purpose, taking the metagenomic assemblies of the 70 samples, we predicted genes and constructed a gene catalogue, a strategy widely exploited for describing the functional potential of microbiomes[61], including both binned and non-binned contigs. We followed a comprehensive gene prediction strategy for accurately predicting both prokaryotic and eukaryotic genes (see Methods), and gathered a total of 25,162,374 genes, to which we assigned functional labels with eggNOG-mapper v2[62]. We then calculated the number of marker genes in each sample, and exploited it for inferring gene copy number per cell, a recommended metric for gene quantification in metagenomes[63,64]. The functional profile of samples from the salinity, heavy metal and 8-factor treatments were similar at general functional categories, and formed a separate cluster from the remaining treatments (Supplementary Fig. 20). However, KEGG orthology (KO) composition of the 8-factor samples was mostly different from the rest, including salinity (Supplementary Fig. 21). For instance, 10 spore germination KOs are significantly depleted in the 8-factor samples, but significantly enriched in the salinity samples with $p < 0.05$ (Two-sided Wilcoxon test).

We next located KEGG pathways with different abundances in the control and 8-factor treatments. Motility and biofilm formation are considered escape mechanisms for some stress conditions[65], but genes related to these processes are significantly depleted when the 8 factors are applied concurrently (Two-sided Wilcoxon test FDR adjusted $q < 0.01$, Fig. 5C). In contrast, these samples show higher

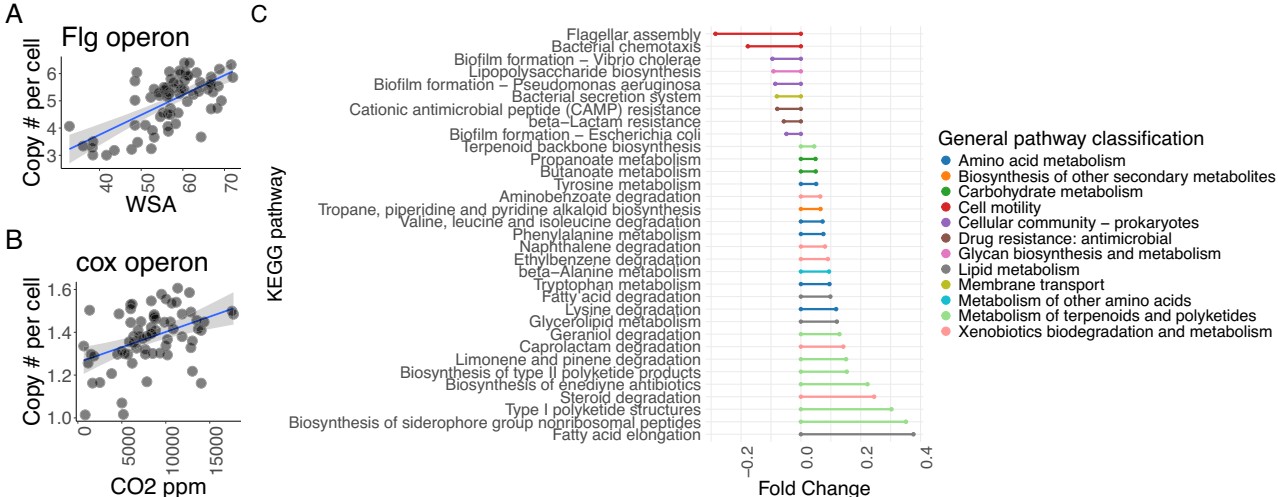

**Fig. 5 | Multiple GC selects for microbial populations with particular gene repertoires. A** Correlation between the frequency of flagellum assembly genes (*flg* operon) and water stable aggregates (WSA). **B** Correlation between the frequency of respiration genes (*coxABCD*) and soil respiration ($CO_2$ ppm). **C** Fold change of frequency of genes involved in several metabolism and degradation

KEGG pathways with significant frequency shifts after the 8-factor treatment compared to control samples (Two-sided Wilcoxon test FDR adjusted $q < 0.01$). In (**A**, **B**), blue lines represent linear regression lines, and shaded areas indicate 95% confidence intervals. Source data are provided as a Source Data file.

pathways (Fig. 5C). These gene content patterns suggest that the 8 GC factors selected for a nutrient recycling life history strategy characterized by increased assimilation and degradation capabilities, instead of an environmental responsiveness strategy[66]. An exception were cytochrome oxidase genes, last enzymatic complexes of most aerobic respiratory chains[67] and markers for respiration, which are significantly depleted after the 8-factor treatment (Two-sided Wilcoxon test, $p = 0.0003$) and significantly correlate (Spearman $R = 0.42$, $p = 0.0003$, Fig. 5B) with $CO_2$ measurements. We explored additional associations between soil properties and gene frequencies and found significant correlations between Water Stable Aggregates (WSA) and genes previously related to soil aggregation (Supplementary Fig. 22). Interestingly, flagellar genes also correlated with WSA (Spearman $R = 0.59$, $p = 9e-10$, Fig. 5A), suggesting a possible relation between bacterial motility and soil aggregation.

**Bacterial composition drives the increase of ARGs in the multifactor samples**

Soil is acknowledged to be an important reservoir of Antimicrobial Resistance Genes (ARGs), some of which are or may become clinically relevant[68]. Hence, it is relevant to monitor ARG distribution across GC treatments for understanding whether they may increase in abundance under different GC conditions. In order to understand changes in ARG frequency, we mapped the genes predicted on each sample against the CARD database[69]. Salinity, heavy metal and 8-factor samples contain higher doses of ARGs, driving a correlation between ARG copy number per cell and community composition ($R = 0.56$, $p = 6e-7$, Fig. 6C) and indicating that phylogeny may be an important driver of ARG frequency, as previously observed in soil[70]. Salinity and the 8 GC factors show a distinctive ARG composition than the other samples (Supplementary Fig. 23), but the multiple concurrent factor treatment drove the strongest increase in both antibiotic inactivation and antibiotic target protection copy number per cell (Two-sided Wilcoxon test, $p = 0.005$ and $p = 0.002$, respectively, Fig. 6A, Supplementary Fig. 24).

We found a strong negative correlation between the copy number of ARGs and the frequency of plasmids (Spearman $R = -0.47$, $p = 2e-5$, Fig. 6D). In fact, only 0.2% ARGs were encoded in contigs classified as plasmids, whereas 1.1% of the genes in the catalogue are. In contrast, we found phage frequency to correlate with ARG copy number

(Spearman $R = 0.31$, $p = 0.006$, Fig. 6E). ARGs were not frequently encoded in phage contigs (0.14% compared to 0.2% in the whole gene catalogue), indicating that phages may not be responsible for the dissemination of these genes, but may indirectly co-select for ARGs.

The increased frequency of ARGs in the 8-factor samples was in part driven by the increase in abundance of *Mycobacterium* species. For instance, the antibiotic target protection increase in frequency was mainly driven by the increase in abundance of the *rbpA* gene (Two-sided Wilcoxon test, $p = 4.33e-5$, Fig. 6B), a RNA-polymerase binding protein which confers resistance to rifampin in *Mycobacterium*[71]. *EfpA*, a MFS transporter typically found in *Mycobacterium tuberculosis* that confers resistance to several antibiotics[72,73], also showed increased frequency after the 8-factor treatment (Two-sided Wilcoxon test, $p = 0.002$). Similarly, the only inactivation gene significantly increasing in frequency after the 8-factor treatment was the metallo-beta lactamase BJP-1, characteristic of *Bradyrhizobium*[74] (Two-sided Wilcoxon test, $p = 0.009$, Fig. 6B).

**Novel gene families distribute differently across taxa and GC treatments**

Because of their unique conditions – e.g., high phylogenetic diversity and number of uncultivated species – soils harbour a large collection of novel genes[61,75], which are acknowledged to play important biological roles[76–78]. Hence, we assessed the distribution of novel genes in microbial populations under different GC conditions. Even though only 14.9% of the genes within the catalogue lack homologs in eggNOG[79], they represent 75% of the gene families built de novo[80] (Fig. 7A).

Only 3.9% novel families are encoded in contigs binned into MAGs, but some bins show high degrees of novel gene content. For instance, more than 20% of the genes from unknown Diplorickettsiaceae family bins and *Alicyclobacillus* genus bins are novel. Additionally, all conditionally rare MAGs contain a higher proportion of novel gene families (e.g., 301 in the genomes enriched after the salinity treatment vs 197 in the rest), some of which may be important to their good performance in different conditions. Even though most novel families (71%) are singletons, some show a wide phylogenetic distribution: 1009 gather more than 100 genes, 748 are binned in MAGs from different bacterial phyla, classes and orders, and 12,139 match novel gene families passing strict quality and evolutionary filters[78].

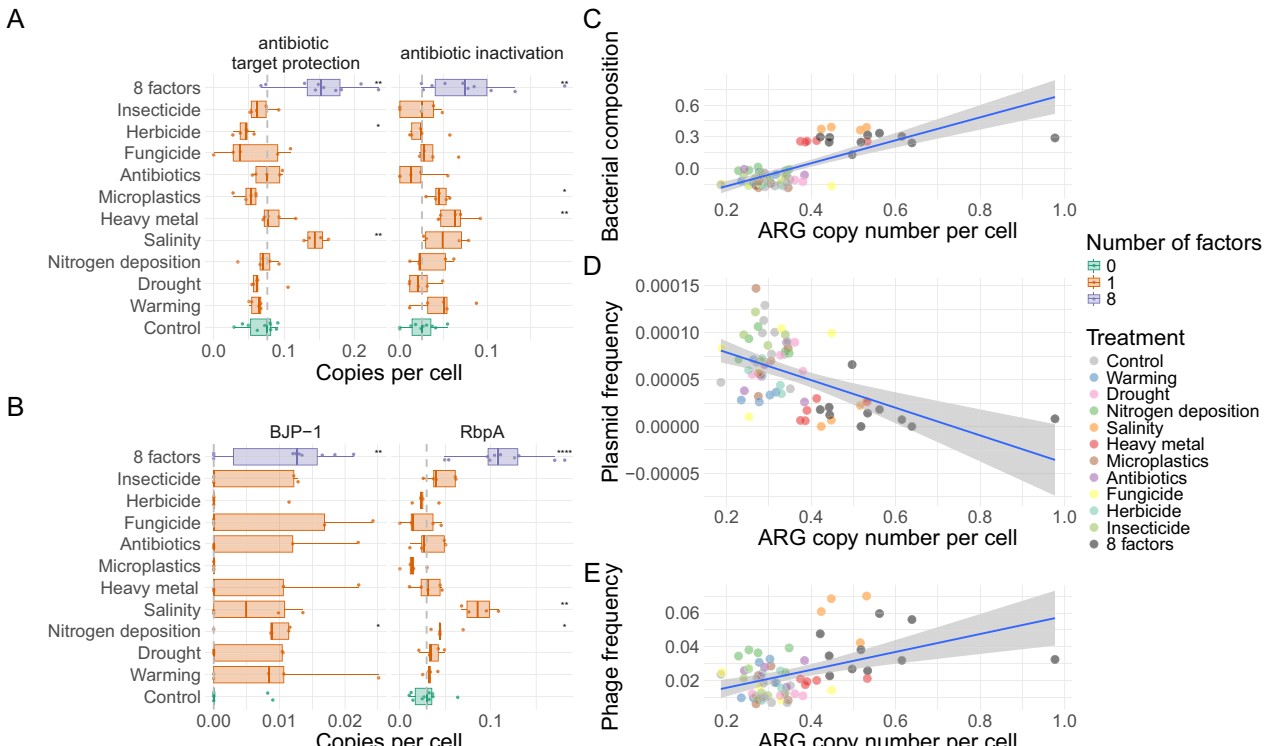

**Fig. 6 | Multiple GC factors drive the increase in abundance of different Antibiotic Resistance Genes (ARGs). A** Antibiotic inactivation and target protection copy number per cell across treatments; (**B**) Copy number per cell variation of *BJP1* (antibiotic inactivation) and *RbpA* (antibiotic target protection). **C**) Correlation between bacterial composition and ARG copy number per cell; (**D**) Correlation between plasmid frequency and ARG copy number per cell. **E**) Correlation between phage frequency and ARG copy number per cell. In (**A**, **B**), data are represented as boxplots in which the middle line is the median, the lower and upper hinges correspond to the first and third quartiles, the upper whisker extends from the hinge to the highest value no further than 1.5 × interquartile range (IQR) from the hinge and the lower whisker extends from the hinge to the lowest value no further than 1.5 × IQR of the hinge. Asterisks represent different significance levels obtained after a Two-sided Wilcoxon test comparing control samples with the samples to which GC treatments were applied, * indicate $p \leq 0.05$, **$p \leq 0.01$, ***$p \leq 0.001$ and ****$p \leq 0.0001$. 10 Control samples, 5 samples for each individual GC treatment, and 10 8-factor samples were considered in the statistical analyses. In (**C**–**E**), blue lines represent linear regression lines, and shaded areas indicate 95% confidence intervals. Exact *p*-values for A and B are provided in Supplementary Tables 9 and 10, respectively. Source data are provided as a Source Data file.

Besides, novel gene families distribute differently across treatments (Fig. 7B), and discriminate the 8-factor samples in a coordinate analysis. For instance, 1163 novel families were exclusively assembled in more than 50% samples from a given treatment, including 64 in antibiotics, possibly representing AMR mechanisms, and 92 in the 8-factor samples. Even though none of these 92 novel gene families were binned into MAGs in our samples, we detected them in other public genomic repositories (GEM, OMD, UHGG, GTDB and GMGC)[16,61,81–83], where 19 distribute across different bacterial phyla and 5 are specific to *Mycobacterium* (one of them to the NTM *Mycobacterium vulneris*[84]). Moreover, we found 10 novel gene families significantly overrepresented (Two-sided Wilcoxon test FDR adjusted $q < 0.05$) in the 8-factor samples, two of them were also present in *Mycobacterium* MAGs. Describing these families remains critical for understanding microbial populations thriving under different GC conditions.

## Discussion

Here we report on the effect of multiple concurrent GC factors on soil bacteria and viruses, complementing earlier results on fungal communities and soil properties[6]. Our study includes 10 individual GC factors, most of which have been broadly studied individually, but not within the same soil context. For instance, salinity has large impacts on bacterial communities, as already noted[85,86], and selects for conditionally rare taxa that are central for a complete understanding of soil's response to GC. Despite the strong effect of salinity, the 8 concurrent factors selected for particular bacterial and viral communities. This is remarkable, as every 8 GC factor sample represents unique conditions which only have the number of factors in common. The different composition of the communities is consistent across different methodologies and biodiversity levels tested (Supplementary Fig. 5). In terms of alpha diversity, MAGs and mOTUs, which provide accurate diversity estimates[26], report diversity losses in the 8-factor treatment. On the contrary, Kraken2, which located a higher number of low-abundant species, revealed an increased diversity after most treatments. Hence, it is possible that the diversity of high abundant species decreases after the 8-factor treatment, but that many rare members become apparent, increasing the overall detected diversity. On the contrary, Kraken2 reports viral diversity losses after the salinity, drought and 8-factor treatment.

A distinctive feature of multiple GC samples is the increased relative abundance of potentially pathogenic *Mycobacterium* genomic bins. Mycobacteria are ubiquitous in soil[25] and known for their ability to survive in harsh conditions[87], which explains their increased abundance in the highly perturbed multifactor samples. These conditions may mirror urban environments, which show more acute global change than natural environments[88], highlighting the possible relevance of this finding to human health. Previous work highlighted that aquatic *Mycobacterium*, more strongly associated with human health, may proliferate under GC conditions[89], but the effect on multiple concurrent factors on aquatic *Mycobacterium* abundance is yet unknown. We could confirm that aquatic *Mycobacterium* increases in relative abundance when exposed to two concurrent GC factors, but additional studies will be needed to confirm this pattern with a higher number of GC factors.

A

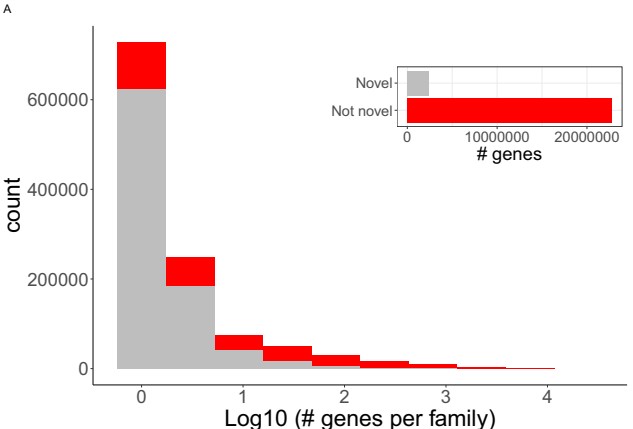

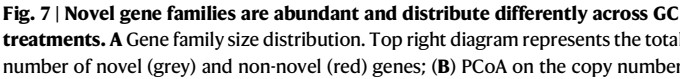

B

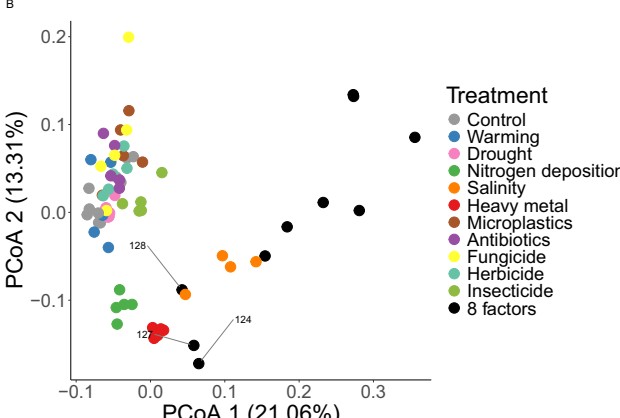

**Fig. 7 | Novel gene families are abundant and distribute differently across GC treatments. A** Gene family size distribution. Top right diagram represents the total number of novel (grey) and non-novel (red) genes; (**B**) PCoA on the copy number per cell of novel gene families assembled in more than 50 samples. We indicate samples 124, 127 and 128, missing the salinity treatment. Source data are provided as a Source Data file.

We also found viral and phage composition to significantly increase in frequency after the application of multiple GC factors, which selected for many unknown taxa. Phages seem to be important in shaping prokaryotic composition, which agrees with their proposed role in regulating host dynamics[12], concretely in highly stressful conditions, which induces prophages to enter the lytic cycle[90].

Changes at the taxonomic level also translated into shifts at the functional level for most treatments, especially salinity, heavy metal and the 8 concurrent factors, which form a separate cluster when considering general functional categories. However, the 8-factor treatment shows a differential composition of KOs, novel gene families and AMR genes, indicating the different metabolism of the microorganisms thriving under these conditions. Given their increased genome size and enriched frequency of metabolic genes, populations surviving multiple GC factors seem to be metabolically diverse, potentially allowing them to leverage a wider range of compounds. In contrast, motility and biofilm formation, which can be resistance mechanisms[91,92], are not selected under multiple GC conditions. Hence, according to our results, under highly perturbed conditions, having wider metabolic capabilities seems to be an important fitness advantage. In fact, given that biomass is not significantly decreased in the 8-factor samples, versatile taxa seem to be able to actively grow, potentially becoming more abundant than populations which prioritize surviving instead of dividing.

We found associations between the frequency of particular genes and previously reported soil processes. For instance, $CO_2$ measurements correlate with the frequency of *cox* genes, central for soil respiration. Similarly, motility genes significantly correlate with WSA, indicating an impact on one another, although in this case the direction of the relation is not straightforward. On the one hand, a meta-analysis suggested that sessile bacteria have a stronger effect on soil aggregation than motile bacteria[93]. On the other hand, motile bacteria may not perform well in low WSA soils because they need water films to swim[94], and such pore spaces may be less optimal in less aggregated soils. Moreover, they may be more exposed to GC factors while moving through soil, decreasing their fitness under multiple GC conditions.

Finally, we also highlight the high abundance of novel gene families in soil samples, and demonstrate that they are relevant for understanding the biology of microbial populations thriving in different GC environments. In fact, even though mostly unbinned, they provide a good discrimination of the 8-factor samples, indicating that uncultivated low-abundant species not captured by MAGs also show a differential response to multiple GC conditions.

Even though our experimental set up was sufficient to demonstrate that multiple GC treatments shape the soil microbiome

differently to any individual treatment, it shows some limitations. By not sequencing additional factor levels (i.e., 2, 5 and 8, as in Rillig et al. (2019)[6]), we could not measure factor interactions, nor test whether prokaryotic and viral community responses scale as the number of concurrent GC factors applied increase, limiting mechanistic interpretations. Additionally, our analysis focuses on a particular time point (6 weeks after treatment) in a grassland soil under highly controlled conditions. Hence, further efforts will be needed to assess whether the strong effect of multiple GC factors is generalizable to less controlled conditions and to alternative soil types, factor number and exposure times. However, this work represents an exhaustive report on the distinctive effect of multiple GC factors on soil prokaryotes and viruses, and urges for incorporating multiple factor treatments when studying the effect of global change.

## Methods
### Experimental design
This study used samples from a controlled environment experiment conducted with 10 factors of global change on soils. Samples from this experiment were immediately frozen at the time of harvest for the analyses described here. For details see Rillig et al. [6]. Briefly, the experiment used mini-bioreactors with 30 g soils to which 10 different GC treatments were applied, either individually, or in combination.

The 10 global change factors of diverse nature considered were i) Warming (increment of 5.0 °C over an ambient temperature of 16.0 °C); ii) Nitrogen enrichment (added the equivalent of 100 kg N ha⁻¹ yr⁻¹ ammonium nitrate to the experimental units in dissolved form); iii) Drought (added half of the amount of water at the beginning of the experiment, compared to control water levels that were at 60% of water holding capacity); iv) Heavy metal (copper (ii)-sulfate–pentahydrate to the soil in dissolved form to a final concentration of 100 mg Cu kg⁻¹); v) microplastics (polyester fibers at a concentration of 0.1%); vi) Salinity (added NaCl to the soil until 4.0 dS m⁻¹); vii) Herbicide (50 mg kg⁻¹ of Roundup® PowerFlex (Monsanto Agrar Deutschland, Düsseldorf), which contains 480 g L⁻¹ glyphosate as active ingredient); viii) Antibiotics (applied oxytetracycline at concentrations of 3.050 mg kg⁻¹); ix) Insecticide (50 ng g⁻¹ of imidacloprid) and x) fungicides (6.0 mg kg⁻¹ of carbendazim). A detailed explanation for the rationale of each treatment can be found at https://www.science.org/doi/suppl/10.1126/science.aay2832/suppl_file/aay2832_rillig_sm.pdf.

For the 8-factor combination, treatments were drawn at random without replacement for each replicate from the set of 10 treatments. This treatment therefore emphasizes the co-occurrence of 8 factors of global change, while de-emphasizing (through the random draws) the

composition of factors. The experiment lasted six weeks, a period sufficient for effects to manifest in such soil experimental systems.

## Shotgun sequencing

Genomic DNA was extracted with the Qiagen DNA Power Soil kit on 250 mg of soil after mixing each sample for homogenization. The genomic DNA was randomly sheared into fragments, which were end repaired, A-tailed and further ligated with Illumina adapters. The fragments with adapters were size selected, PCR amplified, and purified. Sequencing of the 150 bp paired-end reads was performed on an Illumina Novaseq 6000 platform using V1.5 reagent and a S4 flow cell.

## Read processing

Reads obtained from the shotgun metagenome sequencing of the soil samples were trimmed as follows: i) Adapters were removed; ii) Repetitive/overrepresented sequences generated by FASTQC reports were trimmed; iii) Tandem repeats were discarded with the TRF software[95]; iv) Reads were cut when the average quality per base drops below 20, after scanning the read with a 4-base wide sliding window; v) Leading and trailing "*N*" bases or bases with quality lower than 3 were removed.

Reads below 50 bases after trimming, and reads matching to the human genome (hg37dec_v0), were discarded. These filtering steps were run with the kneaddata software (available at https://github.com/biobakery/kneaddata). We rarefied the samples to the library size of the lowest coverage sample using seqtk[96]. Rarefied sequences were only used for quantification purposes.

## Assembly

Contig assembly was performed with SPAdes[97] with the --meta -m 500 --only-assembler parameters. Contigs smaller than 1000 bps were discarded for subsequent analysis.

## Prokaryotic binning

We binned the assembled contigs with: i) SemiBin2[17] with the multi_easy_bin option after mapping the reads to the contigs with BWA[98]; ii) MaxBin 2.0[99] and iii) Metabat2 2[100](-m 1500 -s 100,000 flags). We also merged the predictions of the 3 of them with MAGScoT[101] with default parameters, and with more relaxed options (-t 0.2 -m 10 -a 0.5 -b 0.2 -c 0.2). Genome quality was calculated with CheckM2[102]. SemiBin2 provided the highest number of bins (Supplementary Table 1), which were used for subsequent analysis. We ran dRep[103] for deduplicating the bins and generating representative species bins (95% ANI threshold). We then assigned taxonomic labels to the bins with gtdbtk-2.1.0[19], using the r207 GTDB database as a reference. The relative abundance of each representative bin on each sample was calculated by mapping the rarefied reads against the genomes' contigs with CoverM[104]. For correcting genome size by bin completeness, we divided the MAG size by their completeness value, ranging between 0 and 1 (being 1 a 100% complete MAG). We confirmed our community average genome size estimations with the MicrobeCensus tool[105], ran with default parameters (Supplementary Fig. 2).

For running the phylogenetic tree of the reference MAGs, we first identified marker genes. For that purpose, we mapped the gene predictions of the bins (run with prodigal, see gene prediction section) against the marker genes from the GTDB_r214[81] database. Hits with an e-value < 1e-3 were considered as significant. We then computed gene alignments for each marker gene with MAFFT[106] (--localpair --maxiterate 1000 options), and discarded position with gappiness > 80% with trimAl[107]. The final tree was run with IQ-TREE[108] and -nt AUTO -m GTR + G -cptime 5000 options. For identifying genomes enriched in particular KOs, we used the *mannwhitneyu* python function, and applied the Bonferroni method for correcting for multiple testing.

## Contig classification

We classified contigs into plasmid/chromosomal with i) Plasflow[109] (default options) and ii) PlasmidHunter[110] (default parameters). Contigs predicted as plasmids by the two software were considered to be so. For locating viral contigs, we used VirFinder[57], and considered as viral those contigs with *p* value < 0.05. For locating phages, we ran Seeker[58] on the viral contigs (predict-metagenome script, default options, and considered those contigs with phage probability >0.5 to come from phages). We also ran Whokaryote[111] with default options to identify eukaryotic contigs.

## Viral binning

Viral bins were calculated using the PHAMB software[59], on the bins calculated with VAMB[112], taking the assemblies and the read mappings by BWA used for prokaryotic binning (see above) as input. Viral bin quality was assessed with CheckV[113], and bins labelled as medium quality, high quality and complete were dereplicated with dRep (95% ANI). Viral taxonomy was obtained from the geNomad taxonomy[114] provided by CheckV.

## Gene prediction

For running gene predictions, we ran MetaEuk[115] (easy-predict flag) for the contigs classified as eukaryotic by Whokaryote, and Prodigal[116] (-p meta and -f gff parameters) for the contigs classified as non-eukaryotic. We predicted a low number of genes in eukaryotic contigs (337,262, 1.3% of the total number of genes), as also found in other studies[117]. Eggnog-mapper v2[62] was run for obtaining the functional annotation of the genes (--itype proteins --block_size 0.4 options).

We also identified genes within the CARD[69], mobileOG-db[118] and VFDB[30] databases by mapping the protein sequences with DIAMOND[119] blastp and the sensitive flag. Hits with an *e*-value < 1e$^{-7}$, similarity >80% and coverage >75% were considered as significant[120]. We followed the same approach for locating genes from the VFDB database in public soil MAGs[32].

Flagellar genes within the *flg* operon considered were K02481, K02482, K02386, K02387, K02388, K02389, K02390, K02391. K02392, K02393, K02394, K02395, K02396, K02397, K02398, K02399 (*flgABCDEFGHIJKLMNRS*). Genes within the cytochrome c oxidase cox operon were K02274, K02275, K02276, K02277 (*coxABCD*). WSA related genes considered were K01991 (polysaccharide biosynthesis/export protein, *gfcE*), K09688, K09689 and K10107 (capsular polysaccharide export systems *KpsMTE*), K07091, K09774, K11719, K11720 (lipopolysaccharide export proteins *LptFACG*) and K04077 (H*SP60/GroEL*)[121].

## Gene frequency quantification

Copy number per cell was recently recommended for gene quantification in metagenomes[63,64]. We also decided to use this estimate, instead of relative abundances, because the number of marker genes detected after the different treatments was markedly different (Supplementary Fig. 25), which may be because of a different proportion of eukaryotes or viruses (Fig. S17).

We calculated copy number per cell for each KEGG[122] Orthology, KEGG pathway, CARD and mobile-OG genes. For doing so, we first calculated the mean number of marker genes per sample by mapping the HMMs of the 41 COGs within the fetchMG database (available at http://motu-tool.org/fetchMG.html) with HMMsearch[123] against our gene predictions. Hits with *E*-value < 1e-3 were considered as significant. We then calculated the copy number per cell of a given annotation *X* in a sample *Y* as:

CNPC (*X*,*Y*) = Number of genes homologs to *X* detected in sample *Y*/Average number of marker genes in sample *Y*.

For calculating differentially abundant gene pathways, we computed a Wilcoxon test as implemented in the 'coin' R package, correcting the *p*-values by the FDR method to adjust for multiple testing.

prokaryotic KEGG pathways with $q$-value < 0.01 were considered to be differentially present in control and treated samples.

## Reference-based taxonomic profiling

Given the different performances of alternative taxonomic profiling methods[22], we followed a comprehensive approach, gathering results from different software and reference databases. Taxonomic profiling was performed on the rarefied reads with Kraken2[24], using the k2_pluspf_20230605 database as reference with the --use-names flags. We also computed species abundance by identifying marker genes, regarded to be a more accurate approach[64] with mOTUs[23] v3.1.0 (-t 1 -A -c -q flags) and SingleM[15]. For comparing the taxonomic profiles of the MAG collection and singleM, we recalculated the taxonomic labels of the MAGs with GTDB-tk2 taking the GTDB_r214 as reference, which is the same taken by singleM. For locating pathogenic genera, we looked for the genus set included in the MBPD database[45].

## Species DA analysis on a multiple stressor experiment on water

In order to confirm that *Mycobacterium* species increase in abundance because of multiple factors in water, we downloaded the data generated by Romero et al (2020)[44] from the NCBI (accession number PRJNA574152). We ran the DADA2 pipeline[124] for quantifying the relative abundance of *Mycobacterium* species.

## Novel gene family identification and analysis

We clustered all the gene predictions into gene families with MMseqs2[80] relaxed parameters (--min-seq-id 0.3 -c 0.5 --cov-mode 1 --cluster-mode 2 -e 0.001). We considered gene families with no members detected with eggNOG-mapper 2[62] as novel (i.e. not present in reference species). We located novel families in external MAGs by mapping their longest representative against the proteins encoded in a collection of 169,484 genomes spanning the prokaryotic tree of life (GEM, OMD, UHGG, GTDB and GMGC)[16,61,81–83]. For such a purpose, we used DIAMOND blastp ('sensitive' flag). Hits with an e-value < 1e$10^{-3}$ and query coverage >50% were considered as significant).

## Statistics and figures

Figures were generated with the "ggplot2" R package. Tree figures were generated using the "ggtree" R package. Heatmaps were generated with the *pheatmap* R function. Shannon diversity was calculated with the diversity function within the "microbiome" R package. Beta diversities were calculated with the *vegdist* function within the "vegan" package, using Bray-Curtis distances. Permutations of the multivariate homogeneity of group dispersions (variances) were calculated with the *betadisper* and *permutest* R functions. PCoAs were built using the ape *pcoa* function, providing Bray-Curtis distance matrices computed with the vegan *vegdist* function. The relative importance of each factor was calculated with the *randomForest* R function. In python, we used the "pandas", "scipy" and "numpy" libraries. Sample 85 (salinity treatment) was discarded from the statistical analysis because it shows a high deviation compared to other samples.

## PLFA analysis

For analysis of phospholipid fatty acids (PLFAs), lipids were extracted using a modified Bligh and Dyer method[125,126]. In short, lipids were fractionated into neutral lipids, glycolipids and phospholipids by elution through silica acid columns using chloroform, acetone and methanol, respectively (0.5 g silicic acid, 3 ml; HF BOND ELUT-SI, Varian Inc., Darmstadt, Germany). Phospholipids were subjected to mild alkaline methanolysis and fatty acid methyl esters were identified by chromatographic retention time compared to standards (FAME CRM47885, C11 to C24; BAME 47080-U, C11 to C20; Sigma-Aldrich, Darmstadt, Germany) using a GC-FID Clarus 500 (PerkinElmer Corporation, Norwalk, USA) equipped with an Elite 5 column (30 m × 0.32 mm inner diameter, film thickness 0.25 μm). The temperature programme started with 60 °C

(hold time 1 min) and increased by 30 °C per min to 160 °C, and then by 3 °C per min to 280 °C. The injection temperature was 250 °C and helium was used as carrier gas. Approximately 4 g of fresh soil were used for the extraction.

## Reporting summary

Further information on research design is available in the Nature Portfolio Reporting Summary linked to this article.

## Data availability

The sequencing data generated in this study have been deposited in the NCBI under Bioproject code PRJNA1102178. Prokaryotic and viral bins, and gene quantifications have been deposited in Figshare: https://doi.org/10.6084/m9.figshare.28492940.v1. Amplicon sequencing data from a freshwater GC experiment were retrieved from the NCBI Bioproject PRJNA574152. Source data are provided with this paper.

## Code availability

The code used for the analyses and for generating final figures has been deposited in Github at https://github.com/AlvaroRodriguezDelRio/Multiple_GC_soil_experiment/ and in Zenodo https://doi.org/10.5281/zenodo.14923583. (https://zenodo.org/records/14923583).

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

## Acknowledgements
ARdR was funded by a Humboldt research fellowship for postdoctoral researchers. We thank Anja Wulf for extracting DNA and Stefan Hempel for organizing the sequencing. The authors would like to thank the HPC Service of FUB-IT, Freie Universität Berlin, for computing time (https://doi.org/10.17169/refubium-26754).

## Author contributions
ARdR and MR conceived the project. ARdR performed the metagenomic analyses. SS performed the PLFA analysis. ARdR wrote the manuscript, with comments from all authors.

## Funding

## Competing interests
The authors declare no competing interests.
