## [Transparent Peer Review file · Nature Communications]

Soil microbial responses to multiple global change factors as assessed by metagenomics

Corresponding Author: Dr Alvaro Rodriguez del Rio

Version 0:

Reviewer comments:

Reviewer #1

(Remarks to the Author)

Please, check the attached file with my report.
Thank you

(Remarks on code availability)

Reviewer #2

(Remarks to the Author)

Review of Soil microbial responses to multiple global change factors as assessed by metagenomics by Rodriguez del Rio & Rillig.

The main objective of the research was to determine how ten different global change (GC) factors influenced the bacterial and viral microbial communities. The influence was determined through the analysis of metagenomic data. The authors report changes in bacterial and viral diversity (of assembled MAGs). They use the information derived from MAGs in the GC-treated samples relative to an untreated control to conclude the effect of GC factors on soil microbial communities. One of the noteworthy aspects of the work is that they investigated combined influences of GC parameters.

Major concerns

My main concern relates to the experimental design. The paper investigates 10 GC variables. When investigating these variables in combination, they randomly select 8 at a time. While I appreciate that combinations are investigated, this is a highly unbalanced experimental design. There are numerous methods of balancing experimental variables in orthogonal or orthogonal-like designs. In a balanced design, specific GC effects (or specific combinations of GC effects) can be attributed to specific outcomes with established statistical frameworks. This seems to be an overlooked aspect of the experimental design. In a random design that encompasses 80% of the total variables tested, it's impossible to know which combinations of GC factors are responsible for the observed outcomes. Furthermore, while I appreciated that the variables that included salinity were numbered, in the text, there should be a table showing the specific GC factors for each of the mixed factor samples.

Given the concerns with the unbalanced experimental design, I question how applicable the results are to a larger audience. Or even if the obtained results have relevance outside of this single laboratory mesocosm experiment.

Even with the poor experimental design, the most parsimonious reason the mixed GC treatments often clustered similarly is that the treatments in combination were lethal to many members of the microbial communities. Given that 8/10 variables were combined (and only two were left out), these communities could be experiencing sufficient stress to result in death. Demonstrating this (or not) would significantly strengthen the paper (and make the results clearer).

The observed increase in mycobacterial taxa is consistent with the interpretation of the highly toxic effects of combined treatments. As the authors state, Mycobacteria are highly resistant to many toxins including antibiotics, and heavy metals due to their unusual cell wall construction. Additionally, they are particularly drought resistant. As such, by applying 8 of these factors in combination we might expect to observe a relative enrichment of mycobacteria if their cohabitating microbes are susceptible (i.e. killed) to the treatment.

Viral killing of bacteria might also be important. It's well known that stress induces prophage to enter the lytic cycle of replication and kill their host. Given the changes in the viral communities, this might also be worth exploring.

On a somewhat related note, if cells are dying (which we don't know): 1) relic DNA may become important to remove to get an accurate representation of intact cells, instead of the intact + dead microbial community; and 2) absolute abundances might be more informative when investigating community compositions instead of relative abundances. For example, the increase in the percent of mycobacterial reads could stem from a decrease in the abundance of reads from other taxa (that are potentially dying). qPCR, or some other method of determining absolute changes would resolve this.

Other concerns:

Many of the figures in the main text are illegible because they are too small.

There is quite a bit of literature showing that exposure to soil mycobacteria can have positive effects (for example <https://www.pnas.org/doi/full/10.1073/pnas.1600324113>), so the virulence discussion in around line 160 could be a bit more balanced.

As mentioned above, we should know in the main text what combinations of 8 factors are represented by each 8-factor point.

The sample colors in Fig 1 D and 3 C should be the same given the treatments are the same. Related, are different shapes necessary in the PCA plots?

Are MAG sizes estimates or actual lengths? If actual, they should be estimated lengths based on completeness. If they are estimates, how were they derived?

There are no black bars in Fig. 1 BC as stated in the caption

Perhaps a figure showing the results of the random forest models (change in MSE or something like that) would help

Throughout the paper, Genus names should be italicized.

line 132: MAGs are expected to miss some lineages because they are constructed from an incomplete sampling of the environment. Is there evidence Rhodoplanes were abundant in these samples? It would be nice to have some characterization of the abundant taxa missing from the MAGs.

Lin 278: the 2 in CO₂ is subscripted, not superscripted. Check the entire manuscript.

Fig 4: the colors showing the treatments for ABC are the same colors illustrated in D but mean different things. these should probably be separate figures or there should be a color palette that uses 24 distinct (and interpretable!) colors.

line 293: It's unclear to me why and how the copy number per cell was obtained. A sentence or two explaining that here would help.

line 382: echoing statements above, the survival of mycobacterium in harsh environments could simply be because they are more persistent in an absolute context (ie their numbers are stable, not increasing), and everything else is dying around them. This study can't distinguish these cases as currently designed.

(Remarks on code availability)

Code is legible and available to me. I did not run the code but it appears I could if I wanted to.

Reviewer #3

(Remarks to the Author)

Title: Soil microbial responses to multiple global change factors as assessed by metagenomics

The authors did a controlled experiment to check the effects of 10 global change factors on soil microbial community. They declared that they provided the first report on the effect of multiple concurrent GC factors in soil bacteria and viruses. Yes, I agree this is the first dataset. However, I wonder the confuse of present main objectives. We know these selected factors could reshape soil microbial community, but we want to know more about the processes and the future. Only six weeks incubation can not shape the current situation. So that, I also wonder the applications of present results. Furthermore, the

selected factors are too wide to check the mechanisms. Such as, warming and drought are climate change factors, which display great effects on all ecosystems over the world. In contrast, herbicide and insecticide may show much more effects on farmland rather than natural ecosystems. These 10 factors are not taken place in the same ecosystems. Their processes and aftermath on microbial communities could be obviously different in different soil. The authors can not compare the results among the treatments. Some specific comments as follows,

The abstract should give the reader more specific results, such as the differences among the treatments.

Line 21-24, I don't think the conclusion is acceptable.

Line 46-48, the question is too simple to let us know why care about prokaryotes and viruses.

Line 50, "In order to understand the response of prokaryotes and viruses to multiple GC factors," but the main results mainly describe bacterial community. How about the other prokaryotes?

The section of results is too confused with large information. I suggest to short it and tell us the main novel findings.

The section of discussion likes conclusion. I suggest the authors explain the main results with deep mechanisms or applications.

The methods should give more information about the treatments. The readers may want to see how and why about the treatments.

(Remarks on code availability)

Reviewer #4

(Remarks to the Author)

The manuscript investigated the changes in soil prokaryotic community compositions, functional traits, and viral communities under the stress of global climate (GC) factors using metagenomic analysis, with a focus on the effects of multiple factors.

They found 1. different responses of bacterial and viral communities to various GC factors. 2. Mycobacterium genomic bins were enriched in multiple GC treatment. 3. The changes in ARGs and life history traits under different global change factors provided insights into the interaction between environmental factors and soil microbes.

Major comments

1. The manuscript adopted metagenomic analysis, in particular, genome centric strategy using metagenome bins, however, only 77 MAGs (L78) were obtained for analysis, which may not be able to capture the diversity and shift in bacterial communities.

2. Viral community is targeted, which is understudied. While, soil is one of the richest repositories of viral species diversity. However, the number of viral MAGs obtained is very limited by metagenomic method and may not be representative of the overall changes in the soil viral communities.

3. The samples and experiments of this study come from another published work, the experimental design is crucial for this study. To facilitate reader understanding, it is recommend a more detailed description the experiment.

4. Detail method of the identification of viral contig should be presented. For example, current identification tools heavily relies on the length of contigs. Binning fragmented viral contigs less than 5 kbp may lead to many errors and bacterial contamination. Even though the authors used CheckV for quality filtering, thereby affecting the reliability of diversity assessment. In addition, the majority of the viruses were from the result of Phamb, which is a tool that has not been validated in any environment other than the human gut.

5. One of the conclusion is that multiple GC factors impose directional selective pressure (L21), it is not clear what direction? It seems that multiple GC select for potential pathogens such as mycobacterium (Fig 2), more ARGs (Fig 5), unclassified viruses (figure 3), and functional genes (fig 4). However, it was not clear how other potential pathogens response to GC factors, only a fraction of ARGs (those for antibiotic inactivation and target protection) were presented, similar changes in viral communities and functional genes were observed in single GC treatment (such as salinity treatment). In addition, the number of GC factors is 0, 1 and 8, which may not be able to indicate directional selection.

Some minor comments

1. L74, comparing the genome size of MAGs across different taxa is not meaningful.

2. L81, or maybe due to the quality of MAGs, how about the completeness of represent MAGs?

3. L156-158, it is not surprise to find diverse VFs in mycobacterium bins.

4. L228, Fig S6, the data of two figures seems to be similar.

5. L300-301, it would be better to test the correlation with different types of mobile genes rather than as all.

6. L303-304, other statistical method such as procrustes analysis between bacterial community and ARG profiles should be conducted.

7. Font of figures is too small .

(Remarks on code availability)

Version 1:

Reviewer comments:

Reviewer #2

(Remarks to the Author)

Most of the comments have been addressed. Yet, the explanation for the experimental design remains unsatisfying.

The choice of 8-factor combinations still lacks clear theoretical, empirical, or practical justification. While previous work by the authors showed "ecological surprises" for fungi under 8 randomly drawn global change factors, there's no compelling reason this would extend to prokaryotes and viruses. The omission of intermediate factor levels (2, 5) severely limits our understanding of how community responses scale with disturbance. This design choice appears driven more by convenience than experimental rigor. THE statement in the rebuttal that "the focus was not to understand individual treatments" contradicts basic principles of experimental design. Without systematic analysis of factor interactions, we cannot determine if observed effects stem from specific combinations or truly represent emergent properties of multiple stressors. The statistical framework inadequately addresses these fundamental design limitations. As such, the mechanistic insights gained from this work are consequently limited. A balanced factorial design incorporating intermediate factor levels would have provided much stronger evidence the conclusions while enabling proper analysis of interaction effects.

A few other things from the revision:

line 165: Labeling the "heavy metals" treatment as such is misleading. The treatment is copper specifically and should be labeled and referred to as such. No other metals were studied in the treatment.

Line 267/271: "individually and in combination" and "concurrently" is misleading. These factors were tested together with six other factors. The combination of two factors can't be parsed from the other factors given the experimental design.

(Remarks on code availability)

Reviewer #3

(Remarks to the Author)

The authors addressed most of the comments. I have no more.

(Remarks on code availability)

Reviewer #4

(Remarks to the Author)

The authors have addressed all my concerns and the manuscript has been substantially improved.

(Remarks on code availability)

REVIEWER COMMENTS

Reviewer #1 (Remarks to the Author):

General Comments

This study create a very innovative and challenging effort that pulls together a sort of multiple factors in an attempt to clarify the influence on several Global Warming (GC) measures and their correlations with biodiversity variations. In sequential steps, the authors focused on the response of prokaryotes and viruses to multiple GC factors, by leveraging 70 soil samples from the multi-factor experiment by Rillig et al. (2019), including 10 controls, 50 single GC factor samples (5 samples 52 treated with each individual factor), and 10 multiple GC factor samples (treated with 8 random concurrent GC factors), and analyzed them following a comprehensive metagenomic estrategic coverage. The following step was to reach the unknown species they adopted the method that reveals a great degree of unkown biodiversity by constructing Metagenome-Assembled Genomes (MAGs) de novo. From this step they recovered a total of 742 mostly unknown bacterial and 1,865 viral MAGs from the 70 soil samples, and leveraged them to describe microbial populations under different GC conditions. And as consequence, the authors constructed a 25M gene catalog to monitor changes at the gene functional level, analyzing both known and novel genes. These analyses can bring good insights by indicating that multiple GC factors have a distinctive effect on soil prokaryotic and viral populations and in the distribution of microbial genes. This follow-up study from Rillig et al. (2019) brings several new insights which makes it truly recommended for publication by NatComm, after revision of several points that should be clarified.

We thank the reviewer for their encouraging summary of our work. For clarification, here we focus on Global Change, which gathers Global Warming, as well as other globally occurring, impacts driven by anthropogenic activities on soil ecosystems, like pollutants of different kinds, drought or increased salinity.

Abstract

This section should better explain all the paths taken to indicate why multiple GC factors can impose directional selective pressures on soil prokaryotes and viroses, as one main factor – warming was not the imposing strong effect at functional level (see Fig. S7, line 909).

Thanks for this comment. We now specify in the abstract the analysis we made for demonstrating the differential effect of multiple GC factors on soil prokaryotes and viruses, which was different to that of any individual GC factors, including warming:

- “We recovered a total of 742 mostly unknown bacteria and 1,865 viral MAGs, and leveraged them to describe microbial populations under different GC conditions”
- “We also built a 25M gene catalog to monitor changes at the gene functional level”

We also don't refer to directionality in the abstract anymore, as this cannot be tested with the current experimental set-up.

Also, it is important to describe the multiple 8 GC factors (although tested in random combinations) that imposed directional selective microbial responses to GC against the others 2, 5 and 10 factors. This will attract more attention and could bring a better understanding of the main purpose of this study, initiated by Rillig et al. (2019).

Thanks for pointing this out. We have now incorporated a figure with the random combinations of 8 GC factors (Figure S1), which were different for each sample.

Figure S1

The factor levels 2, 5 and 10 were not included in this study, as we only aimed at demonstrating that applying multiple concurrent GC factors had a contrasting effect on soil microbes than any individual treatment. We have now clarified our analysis by incorporating a graphical summary (Figure 1) and by a more specific framing in the text:

Figure 1

Introduction

The last paragraph presents description of the main steps of the study with data information of the recovery of MAGs from 70 soil samples (no references are shown concerning the type or plant cultivation, as example). The last two phrases look like to be part of the Abstract and not as part of Introduction.

Thanks, we now include the information of the soil type (grassland) in the abstract and introduction. We have also incorporated the information of the main steps taken (previously the last two sentences of the introduction) into the abstract (see comment above).

Results

Why starting referring to “A genome resolved bacterial catalog” instead of describing the initial and necessary steps to recover the bacterial and viral MAGs? I recommend the authors to explain step by step all the pathways taken for this study after Rillig et al. (2019) so the methodology could be followed by others.

We have now started the results section by specifying the steps taken before recovering prokaryotic MAGs from the samples in (Rillig et al. 2019):

- “Hence, after sequencing the metagenomes of the 70 soil samples from the experiment by Rillig et al. (2019), trimming the reads and assembling them into contigs, we aimed at binning them into prokaryotic MAGs”.

We have also clarified the steps taken for building the viral MAGs and the gene catalog in their respective paragraphs:

- “We next exploited the metagenomic assemblies of the 70 samples to ask whether GC conditions also select for different viral populations” and “In order to understand which phage species drove these responses, we computed *de novo* phage bins with PHAMB after MAG calculation with VAMB on the contigs assembled for each sample.”
- “taking the metagenomic assemblies of the 70 samples, we predicted genes and constructed a gene catalog, a strategy widely exploited for describing the functional potential of microbiomes, including both binned and non-binned contigs. We followed a comprehensive gene prediction strategy for accurately predicting both prokaryotic and eukaryotic genes (see methods), and gathered a total of 25,162,374 genes, to which we assigned functional labels with eggNOG-mapper v2. We then calculated the number of marker genes in each sample, and used it for calculating gene copy number per cell, a recommended metric for gene quantification in metagenomes”

Also, the authors should describe the 2, 5, 8 and 10 factors even though they were randomly selected. Also, I do not understand how “warming” was not associated to any of the KEGG pathways (Fig. S7).

As mentioned above, the factor levels 2, 5 and 10 from Rillig et al. (2019) were not included in this study. Also, we are focusing on testing the effect of multiple concurrent GC factors on the soil microbiome, not on the effect of particular ones, like warming, which had a small effect compared to other individual factors like heavy metals or salinity (see for instance beta diversity in Figure 2D, where warming samples cluster closer to control samples).

Please, check additional notes below:

Line 113-114 – Explain what means samples 124, 127 and 128 that were cited so many times throughout the manuscript. What about the others with salinity treatment that were not mentioned?

Thanks for this comment. We indicate these samples because salinity was the main driver of microbial diversity, and the presence or absence of salinity created two clear clusters of 8 GC factor samples. We have clarified this in the text and in the figure legends. We have also included a table indicating the exact GC factor combinations included in every sample (Figure S1).

Figure S1

Line 138-139 – It would be interesting to describe which taxonomic profiling methods were used.

Thanks, we now specify in the main text the taxonomic profiling methods applied (mOTUS (Milanese et al. 2019), SingleM (Woodcroft et al. 2024) and Kraken2 (Wood, Lu, and Langmead 2019)). We have also extended our analysis on these profiles.

Line 140-141 – Indicate what means “read taxonomic classification”.

Thanks, we now changed “read taxonomic classification” to “sequencing read classification based on k-mer frequencies”.

Line 193-194 – If the factor warming was not over all stressfully in this study (Fig. S7, heatmap), explain why the authors showed Fig. S4, from Romero et al. (2020), with the relative abundance of the mycobacterium ASVs located in a freshwater stressor experiment? I could not catch up the idea of including this data, with two factors – warming and pesticides.

The motivation of re-analyzing the data from Romero, Acuña, and Sabater (2020), who performed a multiple stressor experiment on freshwater samples, was to learn whether mycobacteria thrived when multiple factors were applied to freshwater. We found this to be the case in soil, but water is a more important source of Non-Tuberculous Mycobacteria (NMT) infections. The GC factors considered (warming and pesticides) were the ones originally tested in the study, we did not specifically choose them. We have now clarified this in the text.

- “Using data from a previous study by Romero et al. (2020), we tested whether multiple GC factors (warming and pesticides, which were applied individually and in combination in this study) increased the relative abundance of *Mycobacterium* in river biofilms.”

Also, Fig. 4 should be reduced in format.

We have now reduced the size of Figure 4.

Line 236 – increase = increased (correct?)

Thanks, we have now corrected this.

Lines 200 – 202 – Please, clarify as the whole phrase is vaguely written

We have clarified this sentence, which now states:

- “Some of these MAGs were not even detected in control samples, and showed strong increases in abundance after some GC treatments, indicating the power of sample manipulation for uncovering low abundant taxa”

Lines 278, 289, 402 - CO² = CO₂ (correct)

We have now addressed this.

Line 352 - ... “we detected them in other public genomic repositories” – again, too vague.

We now specify the genomic repositories mined (GEM, OMD, UHGG, GTDB and GMGC) in the results and methods section.

Lines 359 – 362 – Why the authors indicate again the samples 124, 127 and 128 in Figure 6?

These 3 samples were treated with 8 GC factors not including the salinity treatment, and form a separate cluster than the rest of the 8 factor samples (see for instance figure 2D). We highlight them for showing how the presence or absence of salinity made an important difference in the 8 factor treatment:

Figure 2D

Discussion

Please, check the notes below:

Line 371-372 – Phrase “Despite...viral communities” – needs correction as it looks like a verb is missing.

We have clarified the sentence, and now state:

- “Despite the strong effect of salinity, the 8 concurrent factors selected for particular bacterial and viral communities.”

Line 372-374 – It is not clear the focus of the statement.

We have clarified the sentence, now:

- “This is remarkable, as every 8 GC factor sample represents unique conditions which only have the number of factors in common”

Methods

I could not correlate the reason why to include a confirmation issue on water as shown in “The sub-item Species DA analysis on a multiple experimente on water” (see lines 562-565), as the present study concentrates on soil samples. I discussed previously this issue when referring to lines 193-194.

We analyzed this data to test whether multiple GC treatments also triggered an increase of mycobacterium relative abundance in freshwater. We did so because our analysis of the soil samples showed that multiple GC factors increased mycobacterial abundance, and wanted to test whether this was the case also in freshwater, which is a more important source of mycobacterial NTM infections.

Reviewer #2 (Remarks to the Author):

Review of Soil microbial responses to multiple global change factors as assessed by metagenomics by Rodriquez del Rio & Rillig.

The main objective of the research was to determine how ten different global change (GC) factors influenced the bacterial and viral microbial communities. The influence was determined through the analysis of metagenomic data. The authors report changes in bacterial and viral diversity (of assembled MAGs). They use the information derived from MAGs in the GC-treated samples relative to an untreated control to conclude the effect of GC factors on soil microbial communities. One of the noteworthy aspects of the work is that they investigated combined influences of GC parameters.

We thank the reviewer for their encouraging summary of our work.

Major concerns

My main concern relates to the experimental design. The paper investigates 10 GC variables. When investigating these variables in combination, they randomly select 8 at a time. While I appreciate that combinations are investigated, this is a highly unbalanced experimental design. There are numerous methods of balancing experimental variables in orthogonal or orthogonal-like designs. In a balanced design, specific GC effects (or specific combinations of GC effects) can be attributed to specific outcomes with established statistical frameworks. This seems to be an overlooked aspect of the experimental design. In a random design that encompasses 80% of the total variables tested, it's impossible to know which combinations of GC factors are responsible for the observed outcomes.

Thanks for this comment. We are aware that having random combinations of 8 GC factors as a multiple GC treatment limits the statistical analysis we could take. For instance, with this non-orthogonal set-up we cannot know the contribution of each GC factor in driving the responses (although the effect of the salinity treatment was evident) or factor interaction types. However, the focus of this work was not to understand the contribution of individual

treatments, but to demonstrate that the combination of a high number of GC factors (regardless of the particular factor combination taken) has a distinctive effect on soil microbial communities. For such a purpose, we consider the random combinations of 8 factors to be optimal, as it represents the multifactorial and variable conditions observed in nature, and which yielded unexpected patterns in the previous study by Rillig et al. (2019). We observed that the random combinations of 8 GC factors consistently selected for particular prokaryotic and viral populations coding for different genes, proving the distinctive effect of having multiple GC factors instead of individual ones, which is the most common approach in GC experiments. In fact, this study represents the first report on the effect of multiple GC factors on soil prokaryotes and viruses, which is relevant even if the response was mostly driven by the interaction of fewer factors. We have now clarified this in the introduction.

Furthermore, while I appreciated that the variables that included salinity were numbered, in the text, there should be a table showing the specific GC factors for each of the mixed factor samples.

Thank you, we agree. We have included a figure indicating which factor combinations were applied to each of the 8 GC factor soil samples (Figure S1).

Figure S1

Given the concerns with the unbalanced experimental design, I question how applicable the results are to a larger audience. Or even if the obtained results have relevance outside of this single laboratory mesocosm experiment.

We believe this setup provides clear evidence that multiple GC factors have a distinctive effect on the microbial communities compared to individual GC treatments. We agree that whether these results hold in other conditions needs to be tested, as we discuss in the manuscript. However, this is the only work testing the effect of multiple GC factors on soil

prokaryotes and so far, and we believe the results are highly relevant in the scope of global change and microbiome research. Also, the microbial results reported in the Rillig et al. (2019) paper have now also been replicated multiple times, e.g. G. Yang et al. (2022), Bi et al. (2024), with different selections of factors and also different soils. We also found the same patterns in the field (unpublished results). We thus believe that this is a much more common phenomenon at least in terms of the basic microbial and soil process patterns.

Even with the poor experimental design, the most parsimonious reason the mixed GC treatments often clustered similarly is that the treatments in combination were lethal to many members of the microbial communities. Given that 8/10 variables were combined (and only two were left out), these communities could be experiencing sufficient stress to result in death. Demonstrating this (or not) would significantly strengthen the paper (and make the results clearer).

Thanks, this is a great suggestion, and we agree that incorporating this information would reinforce the message of the paper. In the view of this, we have conducted a PhosphoLipid Fatty Acid (PLFA) analysis, which indicates the biomass of living cells. We found a variable effect across treatments, with only fungicides significantly shifting bacterial biomass, which may increase because of the reduced fitness of fungal competitors. The concurrent 8 GC factors do not significantly affect bacterial biomass, indicating that bacterial death may only partially drive the response of the soil microbiome to multiple GC factors. We have now incorporated this analysis into the manuscript, and used the PLFA data for calculating absolute abundances (see below).

Figure S9

The observed increase in mycobacterial taxa is consistent with the interpretation of the highly toxic effects of combined treatments. As the authors state, Mycobacteria are highly resistant to many toxins including antibiotics, and heavy metals due to their unusual cell wall

construction. Additionally, they are particularly drought resistant. As such, by applying 8 of these factors in combination we might expect to observe a relative enrichment of mycobacteria if their cohabitating microbes are susceptible (i.e. killed) to the treatment. Viral killing of bacteria might also be important. It's well known that stress induces prophage to enter the lytic cycle of replication and kill their host. Given the changes in the viral communities, this might also be worth exploring.

Thanks, this is a great point. We found that phage frequency strongly correlates with prokaryotic composition, indicating that phages may be important in reshaping the prokaryotic community when experiencing different GC scenarios. We also found a negative but non-significant correlation between phage frequency and bacterial PLFA, suggesting that populations targeted by phages may be substituted by other taxa. We have now incorporated these analysis into the main text.

Figure S18.

On a somewhat related note, if cells are dying (which we don't know): 1) relic DNA may become important to remove to get an accurate representation of intact cells, instead of the intact + dead microbial community; and 2) absolute abundances might be more informative when investigating community compositions instead of relative abundances. For example, the increase in the percent of mycobacterial reads could stem from a decrease in the abundance of reads from other taxa (that are potentially dying). qPCR, or some other method of determining absolute changes would resolve this.

Thanks to the PLFA data, we were able to estimate absolute abundances, which yielded similar community composition estimates (see for instance beta diversity after correcting by bacterial biomass).

Figure S10

Additionally, we have exploited the PLFA data to confirm that, coupled with their elevated relative abundance, mycobacterial biomass increases after the application of 8 concurrent GC factors. We have now incorporated these analyses in the main text.

Figure S11

Other concerns:

Many of the figures in the main text are illegible because they are too small.

We have now increased the font size of all figures.

There is quite a bit of literature showing that exposure to soil mycobacteria can have positive effects (for example <https://www.pnas.org/doi/full/10.1073/pnas.1600324113>), so the virulence discussion in around line 160 could be a bit more balanced.

Thanks, we have now included a note on the positive effect of mycobacteria.

- “Although some mycobacteria have positive health effects, many species, known as Non Tuberculous *Mycobacterium* (NTM) are environmental opportunistic pathogens, and are becoming an increasing sanitary problem”

As mentioned above, we should know in the main text what combinations of 8 factors are represented by each 8-factor point.

We have now incorporated this information in Figure S1.

The sample colors in Fig 1 D and 3 C should be the same given the treatments are the same. Related, are different shapes necessary in the PCA plots?

We now use the same color codes in all figures for indicating treatments and factor levels. We originally included shapes for highlighting the 8 factor samples, but, as this is not needed, we have now reshaped data points in all figures for having more clear images.

Are MAG sizes estimates or actual lengths? If actual, they should be estimated lengths based on completeness. If they are estimates, how were they derived?

Thanks. Previously, MAG sizes were actual lengths, and they should indeed be corrected by completeness. We have now done this by dividing MAG size by their estimated completeness (ranged between 0 and 1). We now specify this in the methods section.

There are no black bars in Fig. 1 BC as stated in the caption

Thanks, we have now addressed this.

Perhaps a figure showing the results of the random forest models (change in MSE or something like that) would help.

Thanks, we have now added this figure to the supplementary material (Figure S4).

Figure S4

Throughout the paper, Genus names should be italicized.

Thanks, we have now addressed this.

line 132: MAGs are expected to miss some lineages because they are constructed from an incomplete sampling of the environment. Is there evidence *Rhodoplanes* were abundant in these samples? It would be nice to have some characterization of the abundant taxa missing from the MAGs.

We agree, this was our motivation to confirm community composition using 3 alternative methods (mOTUS (Milanese et al. 2019), SingleM (Woodcroft et al. 2024) and Kraken2 (Wood, Lu, and Langmead 2019)). We have now extended our analysis of community composition taking the taxonomic profiles calculated by these tools. As suggested by the reviewer, we have identified many taxa missed by the MAG collection, including several archaeal lineages, as well as novel species from the Bacillota phylum not targeted by taxonomic profiling tools (Figure S3). We have also confirmed that *Rhodoplanes* is prevalent in the samples, even though no *Rhodoplanes* MAGs were built.

Figure S3

Lin 278: the 2 in CO₂ is subscripted, not superscripted. Check the entire manuscript.

We have now addressed this, thanks.

Fig 4: the colors showing the treatments for ABC are the same colors illustrated in D but mean different things. these should probably be separate figures or there should be a color palette that uses 24 distinct (and interpretable!) colors.

Thanks, we have now moved the beta diversity panel to the supplementary material and uncolored the points in A and B for having a cleaner figure.

line 293: It's unclear to me why and how the copy number per cell was obtained. A sentence or two explaining that here would help.

We have now added information of how the copy number per cell was calculated in the main text:

- “We then calculated the number of marker genes in each sample, and used it for calculating gene copy number per cell, a recommended metric for gene quantification in metagenomes”

line 382: echoing statements above, the survival of mycobacterium in harsh environments could simply be because they are more persistent in an absolute context (ie their numbers are stable, not increasing), and everything else is dying around them. This study can't distinguish these cases as currently designed.

We have now clarified this with the PLFA data, and confirmed that mycobacterial biomass increases after the application of random combinations of 8 concurrent GC factors (Figure S11).

Figure S11

Reviewer #2 (Remarks on code availability):

Code is legible and available to me. I did not run the code but it appears I could if I wanted to.

Reviewer #3 (Remarks to the Author):

Title: Soil microbial responses to multiple global change factors as assessed by metagenomics

The authors did a controlled experiment to check the effects of 10 global change factors on soil microbial community. They declared that they provided the first report on the effect of

multiple concurrent GC factors in soil bacteria and viruses. Yes, I agree this is the first dataset.

We thank the reviewer for their summary of our work.

However, I wonder the confuse of present main objectives. We know these selected factors could reshape soil microbial community, but we want to know more about the processes and the future. Only six weeks incubation can not shape the current situation. So that, I also wonder the applications of present results.

Microbes are fast in responding to disturbances, and after 6 weeks we already observed strong shifts in microbial communities. We agree that whether these disturbances hold in longer time frames and alternative soil conditions needs to be tested, we have now extended the discussion on this. We also agree that, with the current set up, we cannot know which GC factors are driving the responses in the multiple factor samples, as we also discuss in the paper. However, our goal was to demonstrate the distinctive effect of multiple concurrent GC factors on soil microbes, not to test which factor combinations were more relevant. For this, we used random combinations of 8 GC factors, which represents the multifactorial and changing conditions observed in nature and whose effect has never been measured on soil prokaryotes. In fact, this is the only study so far on the effect of multiple concurrent GC factors in soil prokaryotes, viruses and their gene functional repertoire, and we believe the results are highly relevant in the scope of global change and microbiome research.

Additionally, the microbial results reported in the Rillig et al. (2019) paper have now also been replicated multiple times, e.g. G. Yang et al. (2022) and Bi et al. (2024), with different selections of factors and also different soils. We also found the same patterns in the field (unpublished results). We thus believe that this is a much more common phenomenon at least in terms of the basic microbial and soil process patterns.

Furthermore, the selected factors are too wide to check the mechanisms. Such as, warming and drought are climate change factors, which display great effects on all ecosystems over the world. In contrast, herbicide and insecticide may show much more effects on farmland rather than natural ecosystems.

We agree that some of the global change factors tested have a wider distribution than others. However, these are all globally occurring factors triggered by anthropogenic activities and affecting biota, meeting the global change factor definition. The definition of global change means that a factor has to have global scope, but no global change factor is active or equally active everywhere on Earth.

These 10 factors are not taken place in the same ecosystems. Their processes and aftermath on microbial communities could be obviously different in different soil. The authors can not compare the results among the treatments.

Many of these factors are expected to co-occur in urban and many agricultural environments, and in areas affected by off-target effects (we used concentrations to mimic those). However, the aim of this work is not to test the effect of factor combinations that happen in particular terrestrial ecosystems, but to show how applying a high number of GC

factors has a different effect than applying GC factors individually, which is the most commonly applied focus in GC research on soil (in the Rillig et al. 2019 paper we show that 98.2% of all papers used only 1 or 2 factors when experimentally examining global change effects on soils).

Some specific comments as follows,

The abstract should give the reader more specific results, such as the differences among the treatments.

We have now added additional information of the experimental design and specific results in the abstract:

- "Here, we applied 10 GC treatments (warming , drought, nitrogen deposition, increased salinity, heavy metals, microplastics, antibiotics, fungicides, herbicides and insecticides) individually and in random combinations of 8 factors to soil grassland samples"
- "We recovered a total of 742 mostly unknown bacteria and 1,865 viral MAGs"
- "We also built a 25M gene catalog to monitor changes at the gene functional level"

Line 21-24, I don't think the conclusion is acceptable.

It is true that directionality cannot be derived from our data, and have deleted this from the concluding sentence in the abstract. However, we believe that the conclusion (that we demonstrate that multiple GC factors have differential effects than individual ones) is fully supported by the data, and we hope this is more clear now after the edits to other parts of the text.

Line 46-48, the question is too simple to let us know why care about prokaryotes and viruses.

Thanks, we have now extended our explanation on the importance of prokaryotes and viruses on soil functioning in the introduction:

- "This includes prokaryotes, which usually show different dynamics than fungi and are central to soil functioning. For instance, they are the only fixers of molecular nitrogen, a limiting soil nutrient, mediate phosphorus mobilization, critical for plant growth· decompose plant derived organic matter, and contribute to soil structure through the formation of aggregates. Additionally, the response of viruses, understudied players of soil functioning with a key role in regulating microbial host dynamics and soil carbon pools, to multiple GC factors has also not been studied."

Line 50, "In order to understand the response of prokaryotes and viruses to multiple GC factors," but the main results mainly describe bacterial community. How about the other prokaryotes?

We previously focused on bacteria because all the MAGs we recovered were bacterial. However, we have now incorporated additional analysis taking taxonomic profiles based on reference databases, which were able to locate archaeal taxa (e.g. see Figure S3 for a

comparison of the taxa included in our MAGs and taxa detected by SingleM, a reference based taxonomic profiling tool that locates several archaeal lineages).

Figure S3

The section of results is too confused with large information. I suggest to short it and tell us the main novel findings.

We are aware that the results section is quite long, but we consider all the information provided to be relevant and needed to convey the essence of the findings.

The section of discussion likes conclusion. I suggest the authors explain the main results with deep mechanisms or applications.

Thanks, we have now extended the conclusion section with additional interpretations of the data.

The methods should give more information about the treatments. The readers may want to see how and why about the treatments.

We have now added a link to a previously published report explaining the rationale of the different treatments in the methods section.

Reviewer #4 (Remarks to the Author):

The manuscript investigated the changes in soil prokaryotic community compositions, functional traits, and viral communities under the stress of global climate (GC) factors using metagenomic analysis, with a focus on the effects of multiple factors. They found 1. different responses of bacterial and viral communities to various GC factors. 2. Mycobacterium

genomic bins were enriched in multiple GC treatment. 3. The changes in ARGs and life history traits under different global change factors provided insights into the interaction between environmental factors and soil microbes.

We thank the reviewer for their encouraging summary of our work.

Major comments

1. The manuscript adopted metagenomic analysis, in particular, genome centric strategy using metagenome bins, however, only 77 MAGs (L78) were obtained for analysis, which may not be able to capture the diversity and shift in bacterial communities.

Thanks, we have now expanded the prokaryotic and viral community analyses with additional taxonomic profiles based on mappings to reference databases (mOTUS (Milanese et al. 2019), SingleM (Woodcroft et al. 2024) and Kraken2 (Wood, Lu, and Langmead 2019)), which located a higher number of species than the *de novo* MAG collection. The patterns we initially observed are robust across methodologies (except for alpha diversity, which varied across methods), reinforcing our results.

2. Viral community is targeted, which is understudied. While, soil is one of the richest repositories of viral species diversity. However, the number of viral MAGs obtained is very limited by metagenomic method and may not be representative of the overall changes in the soil viral communities.

Thanks, we have now incorporated an additional analysis on the viral communities taking the taxonomic profiles by Kraken2 (see Figure S19).

3. The samples and experiments of this study come from another published work, the experimental design is crucial for this study. To facilitate reader understanding, it is recommend a more detailed description the experiment.

Thanks, we have added additional information in the abstract and introduction, and included a summary figure (Figure 1) to clarify the experimental setup:

Figure 1

4. Detail method of the identification of viral contig should be presented. For example, current identification tools heavily relies on the length of contigs. Binning fragmented viral contigs less than 5 kbp may lead to many errors and bacterial contamination. Even though the authors used CheckV for quality filtering, thereby affecting the reliability of diversity assessment. In addition, the majority of the viruses were from the result of Phamb, which is a tool that has not been validated in any environment other than the human gut.

In the first version of the manuscript, we only used the viral MAGs from PHAMB (Johansen et al. 2022) for exploring viral community composition. Because of the concerns raised by the reviewer, and for expanding the taxonomic profiles to non-phage viruses, we have now incorporated an additional analysis on the viral community composition based on the results by Kraken2. We have also clarified how the viral MAGs were constructed in the results:

- "In order to understand which phage species drove these responses, we computed *de novo* phage bins with PHAMB after MAG calculation with VAMB on the contigs assembled for each sample"

and added more detailed information on how we computed viral MAGs in the methods section:

- "Viral bins were calculated using the PHAMB software, on the bins calculated with VAMB, taking the assemblies and the read mappings by BWA used for prokaryotic binning (see above) as input"

5. One of the conclusion is that multiple GC factors impose directional selective pressure (L21), it is not clear what direction? It seems that multiple GC select for potential pathogens such as mycobacterium (Fig 2), more ARGs (Fig 5), unclassified viruses (figure 3), and functional genes (fig 4). However, it was not clear how other potential pathogens response to GC factors, only a fraction of ARGs (those for antibiotic inactivation and target protection) were presented, similar changes in viral communities and functional genes were observed in single GC treatment (such as salinity treatment). In addition, the number of GC factors is 0, 1 and 8, which may not be able to indicate directional selection.

We have now addressed this and do not state that multiple GC factors drive directional responses, as directionality cannot be demonstrated with our data, only that multiple GC factors have a particular effect different from any individual treatment. Even though salinity is driving most of the response, we demonstrate that the 8 factor samples have different characteristics than the salinity samples in terms of bacterial and viral composition and gene repertoire. This is well represented In all beta diversity analysis, in which salinity and 8 factor samples cluster in two different groups along the second PCoA axis, which usually explain a high degree of variability observed (e.g. always higher than 10%). Some particular differences between the salinity and 8 factor samples include Firmicutes abundance (Figure 2A) or spore germination gene copy number per cell, which show contrasting patterns in the two treatments.

Figure 2A

In the previous version, we only targeted mycobacteria as potential pathogens because no other potentially pathogenic taxa was found among the MAGs. We have now monitored the abundance of additional genera containing pathogenic species (compiled in the MBPD database by X. Yang et al. (2023)) in the taxonomic profiles calculated by singleM (Figure S15). Apart from mycobacteria, we found additional genera including pathogenic species increasing in abundance after the 8 factor treatment, like *Bacillus*, *Paenibacillus* and *Psychrobacillus*.

Figure S15

We have also included the frequency of all the CARD (Alcock et al. 2020) ARG categories into Figure S24. We initially decided not to report the results on the “antibiotic target

alteration” categories, which also show significant changes, because these may be gene variants with high degree of similarity which we cannot distinguish by the identity thresholds applied. We also did not show the antibiotic efflux and reduced permeability categories because no significant changes after any of the treatments were observed.

Figure S24

Some minor comments

1.L74, comparing the genome size of MAGs across different taxa is not meaningful.

Genome size is an important indicator of life history strategies in terrestrial microbes, and, as MAGs usually represent the most abundant taxa, we believe MAG size is a good proxy for community average genome length. However, we have now confirmed these results with the MicrobeCensus tool (Nayfach and Pollard, 2015), aimed at estimating the average genome size of a microbial community. The average genome size calculated by MicrobeCensus are generally higher than MAG size, even after correcting by MAG completeness, but both strongly correlate ($R = 0.56, p\text{-value} = 5e-7$), and yield similar patterns. We have now

incorporated these analyses in the main text, and incorporated Figure S3:

Figure S3

2.L81, or maybe due to the quality of MAGs, how about the completeness of represent MAGs?

We have now corrected genome size by MAG completeness.

3.L156-158, it is not surprise to find diverse VFs in mycobacterium bins.

Thanks, we have now clarified this in the text:

- “In order to understand the potential pathogenicity of the unknown *Mycobacterium* MAGs thriving in multiple GC conditions, we examined their virulence factor content, which, despite not being markers for pathogenicity, contribute to the ability of pathogens to survive within hosts, and are common in mycobacteria”

4.L228, Fig S6, the data of two figures seems to be similar.

The data shown in Fig. S6A and S6B (now S17) show similar patterns, which is not surprising because (A) shows the proportion of viral contigs, and (B) the proportion of phage contigs, which gather most of the viral contigs detected.

5.L300-301, it would be better to test the correlation with different types of mobile genes rather than as all.

Thanks, this was a great point. After considering plasmids and phages separately, instead of looking at overall MGEs, we found that ARGs negatively correlate with plasmid abundance, indicating that plasmids may have a limited role in ARG spread in the samples. On the contrary, we found that ARG frequency significantly correlates with phage abundance. Because we did not find a particularly high number of genes in phage contigs, we believe that phages may not be promoting ARG spread by actively carrying ARG genes, but by

indirectly selecting for antibiotic resistant species. We have now incorporated this analysis in the main text and figures.

Figure 6

6.L303-304, other statistical method such as procrustes analysis between bacterial community and ARG profiles should be conducted.

We have now calculated beta diversities on the abundance profiles of ARGs. Salinity and the 8 factor samples form a clear cluster, indicating that salinity is driving the response, even though some differences across categories in the two treatments exist (see Figure 6). We have now included this analysis in the main text.

7. Font of figures is too small .

We have now increased the font size of all figures.

References

- Alcock, Brian P., Amogelang R. Raphenya, Tammy T. Y. Lau, Kara K. Tsang, Mégane Bouchard, Arman Edalatmand, William Huynh, et al. 2020. "CARD 2020: Antibiotic Resistome Surveillance with the Comprehensive Antibiotic Resistance Database." *Nucleic Acids Research* 48 (D1): D517–25.
- Bi, Mohan, Huiying Li, Peter Meidl, Yanjie Zhu, Masahiro Ryo, and Matthias C. Rillig. 2024. "Number and Dissimilarity of Global Change Factors Influences Soil Properties and Functions." *Nature Communications* 15 (1): 8188.
- Johansen, Joachim, Damian R. Plichta, Jakob Nybo Nissen, Marie Louise Jespersen, Shiraz A. Shah, Ling Deng, Jakob Stokholm, et al. 2022. "Genome Binning of Viral Entities from Bulk Metagenomics Data." *Nature Communications* 13 (1): 965.
- Milanese, Alessio, Daniel R. Mende, Lucas Paoli, Guillem Salazar, Hans-Joachim Ruscheweyh, Miguelangel Cuenca, Pascal Hingamp, et al. 2019. "Microbial Abundance, Activity and Population Genomic Profiling with mOTUs2." *Nature Communications* 10 (1): 1014.
- Nayfach, S. & Pollard, K. S. Average genome size estimation improves comparative metagenomics and sheds light on the functional ecology of the human microbiome. *Genome Biol.* 16, 51 (2015).
- Rillig, Matthias C., Masahiro Ryo, Anika Lehmann, Carlos A. Aguilar-Trigueros, Sabine Buchert, Anja Wulf, Aiko Iwasaki, Julien Roy, and Gaowen Yang. 2019. "The Role of Multiple Global Change Factors in Driving Soil Functions and Microbial Biodiversity." *Science* 366 (6467): 886–90.
- Romero, Ferran, Vicenç Acuña, and Sergi Sabater. 2020. "Multiple Stressors Determine Community Structure and Estimated Function of River Biofilm Bacteria." *Applied and*

- Environmental Microbiology* 86 (12). <https://doi.org/10.1128/AEM.00291-20>.
- Woodcroft, Ben J., Samuel T. N. Aroney, Rossen Zhao, Mitchell Cunningham, Joshua A. M. Mitchell, Linda Blackall, and Gene W. Tyson. 2024. "SingleM and Sandpiper: Robust Microbial Taxonomic Profiles from Metagenomic Data." *bioRxiv*. <https://doi.org/10.1101/2024.01.30.578060>.
- Wood, Derrick E., Jennifer Lu, and Ben Langmead. 2019. "Improved Metagenomic Analysis with Kraken 2." *Genome Biology* 20 (1): 257.
- Yang, Gaowen, Masahiro Ryo, Julien Roy, Daniel R. Lammel, Max-Bernhard Ballhausen, Xin Jing, Xuefeng Zhu, and Matthias C. Rillig. 2022. "Multiple Anthropogenic Pressures Eliminate the Effects of Soil Microbial Diversity on Ecosystem Functions in Experimental Microcosms." *Nature Communications* 13 (1): 4260.
- Yang, Xinrun, Gaofei Jiang, Yaozhong Zhang, Ningqi Wang, Yuling Zhang, Xiaofang Wang, Fang-Jie Zhao, Yangchun Xu, Qirong Shen, and Zhong Wei. 2023. "MBPD: A Multiple Bacterial Pathogen Detection Pipeline for One Health Practices." *iMeta* 2 (1): e82.

Reviewer #2 (Remarks to the Author):

Most of the comments have been addressed.

We thank the reviewer for all their comments, which have improved the quality of the paper.

Yet, the explanation for the experimental design remains unsatisfying. The choice of 8-factor combinations still lacks clear theoretical, empirical, or practical justification. While previous work by the authors showed "ecological surprises" for fungi under 8 randomly drawn global change factors, there's no compelling reason this would extend to prokaryotes and viruses. The omission of intermediate factor levels (2, 5) severely limits our understanding of how community responses scale with disturbance. This design choice appears driven more by convenience than experimental rigor. THE statement in the rebuttal that "the focus was not to understand individual treatments" contradicts basic principles of experimental design. Without systematic analysis of factor interactions, we cannot determine if observed effects stem from specific combinations or truly represent emergent properties of multiple stressors. The statistical framework inadequately addresses these fundamental design limitations. As such, the mechanistic insights gained from this work are consequently limited. A balanced factorial design incorporating intermediate factor levels would have provided much stronger evidence the conclusions while enabling proper analysis of interaction effects.

We thank the reviewer for highlighting the limitations of our experimental design. Even though it confidently shows that multiple GC factors shape prokaryotic and viral communities differently than any individual GC factor, we agree that mechanistic interpretations are limited. We have now included a paragraph explaining the limitations of our experimental design in the Discussion section. We note that this experimental design involving random draws does not permit the study of individual interactions, as has already been made clear in previous work (e.g. Rillig et al. 2019, Science; Bi et al. 2024, Nature Communications).

A few other things from the revision:

line 165: Labeling the "heavy metals" treatment as such is misleading. The treatment is copper specifically and should be labeled and referred to as such. No other metals were studied in the treatment.

Thanks. We do see the reviewer's point. However, copper was applied as an example of a heavy metal, just like the other substances are examples of their respective category. Changing the label to copper would be inconsistent with the other treatment labels.

Line 267/271: "individually and in combination" and "concurrently" is misleading. These factors were tested together with six other factors. The combination of two factors can't be parsed from the other factors given the experimental design.

In this paragraph, we present the results of re-analyzing data from a previous study (Romero et al., 2020), in which they tested the effect of warming and pesticides (individually and in combination) on the microbiome of river biofilms.

Reviewer #3 (Remarks to the Author):

The authors addressed most of the comments. I have no more.

We thank the reviewer for their comments, which have improved the quality of the paper.

Reviewer #4 (Remarks to the Author):

The authors have addressed all my concerns and the manuscript has been substantially improved.

We thank the reviewer for their comments, which have improved the quality of the paper.

REVIEWER'S SUGGESTIONS

Soil microbial responses to multiple global change factors as assessed by metagenomics

Authors: Álvaro Rodríguez del Río, Matthias C. Rillig

General Comments

This study create a very innovative and challenging effort that pulls together a sort of multiple factors in an attempt to clarify the influence on several Global Warming (GC) measures and their correlations with biodiversity variations. In sequential steps, the authors focused on the response of prokaryotes and viruses to multiple GC factors, by leveraging 70 soil samples from the multi-factor experiment by Rillig *et al.* (2019), including 10 controls, 50 single GC factor samples (5 samples 52 treated with each individual factor), and 10 multiple GC factor samples (treated with 8 random concurrent GC factors), and analyzed them following a comprehensive metagenomic estrategic coverage. The following step was to reach the unknown species they adopted the method that reveals a great degree of unkown biodiversity by constructing Metagenome-Assembled Genomes (MAGs) *de novo*. From this step they recovered a total of 742 mostly unknown bacterial and 1,865 viral MAGs from the 70 soil samples, and leveraged them to describe microbial populations under different GC conditions. And as consequence, the authors constructed a 25M gene catalog to monitor changes at the gene functional level, analyzing both known and novel genes. These analyses can bring good insights by indicating that multiple GC factors have a distinctive effect on soil prokaryotic and viral populations and in the distribution of microbial genes. This follow-up study from Rillig *et al.* (2019) brings several new insights which makes it truly recommended for publication by NatComm, after revision of several points that should be clarified

Abstract

This section should better explain all the paths taken to indicate why multiple GC factors can impose directional selective pressures on soil prokaryotes and viroses, as one main factor – warming was not the imposing strong effect at functional level (see Fig. S7, line 909). Also, it is important to describe the multiple 8 GC factors (although tested in random combinations) that imposed directional selective microbial responses to GC against the others 2, 5 and 10 factors. This will attract more attention and could bring a better understanding of the main purpose of this study, initiated by Rillig *et al.* (2019).

Introduction

The last paragraph presents description of the main steps of the study with data information of the recovery of MAGs from 70 soil samples (no references are shown concerning the type or plant cultivation, as example). The last two phrases look like to be part of the Abstract and not as part of Introduction.

Results

Why starting referring to “A genome resolved bacterial catalog” instead of describing the initial and necessary steps to recover the bacterial and viral MAGs? I recommend the authors to explain step by step all the pathways taken for this study after Rillig *et al.* (2019) so the methodology could be followed by others.

Also, the authors should describe the 2, 5, 8 and 10 factors even though they were randomly selected.

Also, I do not understand how “warming” was not associated to any of the KEGG pathways (Fig. S7).

Please, check additional notes below:

Line 113-114 – Explain what means samples 124, 127 and 128 that were cited so many times throughout the manuscript. What about the others with salinity treatment that were not mentioned?

Line 138-139 – It would be interesting to describe which taxonomic profiling methods were used.

Line 140-141 – Indicate what means “read taxonomic classification”.

Line 193-194 – If the factor warming was not over all stressfully in this study (Fig. S7, heatmap), explain why the authors showed Fig. S4, from Romero et al. (2020), with the relative abundance of the mycobacterium ASVs located in a freshwater stressor experiment? I could not catch up the idea of including this data, with two factors – warming and pesticides. Also, Fig. 4 should be reduced in format.

Line 236 – increase = increased (correct?)

Lines 200 – 202 – Please, clarify as the whole phrase is vaguely written

Lines 278, 289, 402 - $CO^2 = CO_2$ (correct)

Line 352 - ... “we detected them in other public genomic repositories” – again, too vague.

Lines 359 – 362 – Why the authors indicate again the samples 124, 127 and 128 in Figure 6?

Discussion

Please, check the notes below:

Line 371-372 – Phrase “Despite...viral communities” – needs correction as it looks like a verb is missing.

Line 372-374 – It is not clear the focus of the statement.

Methods

I could not correlate the reason why to include a confirmation issue on water as shown in “The sub-item Species DA analysis on a multiple experimente on water” (see lines 562-565), as the present study concentrates on soil samples. I discussed previously this issue when referring to lines 193-194.